# Sustainability of religious communities

**Chulsu Jo**[1☯], **Doo Hwan Kim**[2,3☯], **Jae Woo Lee**[3☯]*

**1** Future Changes Forecasting Institute, McAllen, TX, United States of America, **2** Future Changes Forecasting Institute, Incheon, South Korea, **3** Department of Physics, Inha University, Incheon, South Korea

☯ These authors contributed equally to this work.
* jaewlee@inha.ac.kr

## Abstract

This article focuses on the relationship between church population and sustainability. We carried out the study on a sample of Presbyterian churches in South Korea, and implemented dynamic optimization of the church population based on the Susceptible–Infected–Recovered (SIR) epidemic model. In particular, System Dynamics (SD) and Agent-Based Model (ABM) simulations are performed for a prototype model with key parameters that contribute to church growth. Potential parameters reflecting sustainability for churches trigger dramatic growth in church populations. We categorized five dimensions of sustainability with various multi-dimensional indicators in order to measure the level of sustainability, and we obtained the values of the indicators by analyzing a number of news articles searched with a text mining technique. As time-dependent values of sustainability are imposed on the generic SD model for church population dynamics as sustainable potential parameters, the optimized result reproduces specific features for the church population. We discuss the roles of key parameters for sustainable church growth, and the contributions of the churches to sustainability.

**Data Availability Statement:** All relevant data are within the paper and its Supporting Information files.

**Funding:** JW was supported by the Basic Science Research Program through the National Research

## Introduction

Religion can be considered a fertile source of social values for sustainability [1]. The fields addressing the relationship between religion and sustainability have been studied, referring to the potential role of religion in sustainability transitions [2, 3]. Sustainability transition is a long-term, multi-dimensional, and fundamental transformation process, which establishes that a socio-technical system shifts more toward sustainable modes of production and consumption [4]. The sociological aspects of religion include institutional structure, spirituality, leadership and membership, worldview, and religious routines and populations, which can provide diversity to the types of niches that enhance sustainability transitions [5]. Religious beliefs could provide a pragmatic force to create purposeful transitions by sharing ecologically positive habits of practice and attitudes with succeeding generations, working toward a greener future [6, 7]. Michael Ben-Eli suggested a holistic definition of sustainability, including spiritual domain [8]. Discourse on religion for sustainability is related to the spiritual domain in which the correlation patterns provide understanding of the functions of religion.

Michael Ben-Eli asserted that most of the existing tools, concepts, institutions, frameworks, and mechanisms are not adequate at addressing sustainability for a huge population and the

Foundation of Korea (Grant No. NRF-
2020R1A2C1005334).

**Competing interests:** The authors have declared
that no competing interests exist.

urges to evolve in order to accommodate an entirely different reality [8]. According to Lior
et al., an advancing sustainability science should strengthen its empirical, theoretical, and prac-
tical fundamentals focused on the role of values in the field of science and decision-making for
sustainability [9]. Thus, theoretical and empirical research should note that the complex inter-
play of religion and sustainability cannot be expressed by simply equating religious beliefs with
sets of uncertain values [1]. Therefore, the relationship between religion and sustainability
may be accounted for by analyses of measurable, sustainable factors reflecting a sustainability
effect in the population of a religious community.

Religious communities carry out a variety of methods with the aim of maintaining their
own sustainability. The United Methodist Church used religious branding to enhance the pub-
lic reputation and to change the position of the traditional denomination. It demonstrated the
need of marketing tool for the sustainability of religious institutions [10]. Oosthuizen and
Lategan pointed out that church leaders often had difficulty performing the basic management
tasks they expected because leaders had insufficient management principles and skills as an
organization [11]. The authors argued that church and denomination leaders should partici-
pate in management principles and skills education for more effective, efficient and sustainable
management. Körösvölgyi addressed that as the Christian population declined, it was neces-
sary to look at the positive and negative experiences affecting the sustainability of the church
amid changes in the world map such as global warming, urbanization, uneven distribution of
population and wealth, and migration [12]. The author argued that if the place of worship as
sacred building equipped with an efficient architectural concept to be better suited to the com-
munity, it could have a positive impact on the sustainability of the church.

Church planting is a good strategy if the church is to be sustainable through conversion.
Churches wishing to grow should invest equally in member satisfaction and recruitment (con-
version). In addition, the ministry of welcoming new believers and discipling them should be
balanced [13]. De Wetter and Roozen described the health and growth of the church as the
concept of church vitality [14, 15]. They insisted that the spiritual vitality of worship through
enthusiastic, committed or active members could be the catalyst for membership growth.

Chai showed through the small group ministry called as the house church in the Seoul Bap-
tist Church of Houston that the growth and sustainability of the church could be caused by
conversion of non-believers [16]. By converting the traditional church into a small group-cen-
tered church, the church focused on converting non-believers and raising beginners as disci-
ples. Churches participating in house church ministry reeducated pastors and lay leaders by
holding seminars and conferences after establishing the institute named as The House Church
Ministries with Chai. The Seoul Baptist Church of Houston also supported that other churches
could sustain the small group ministry by providing short-term training.

Church enthusiasts lead to successful evangelism by adding new members when they are
primarily involved in the conversion of unbelievers. It is the sustainability of the church that
drives growth through its ability to reproduce itself [17, 18]. If the new believer's faith is not
strengthened, they will lose their passion for faith after a certain period of time. New believers
are more likely to lose networks with unbelieving friends, and they will have a hard time form-
ing new networks within the church. Therefore, sustainability potential represents the degree
to which unbelievers contact believers and form trust. It represents the degree of trade-off
between what unbelievers lose as they gain faith and what they gain in social networks and
new communities.

Hayward studied church growth using mathematical population modeling using the System
Dynamics (SD) approach based on the Susceptible–Infected–Recovered (SIR) epidemic
model, which proposed a diffusive mechanism of the church population from contact between
unbelievers and believers with enthusiasm [19, 20]. Madubueze and Nwaokolo [21] and

Ochoche and Gweryina [22] numerically solved differential model equations for church populations based on epidemiological modeling, and evaluated the impact of active believers. McCartney and Glass constructed a set of models with coupled differential equations using a three-state dynamic model for religious affiliations [23]. Chattoe formalized the Agent-Based Model (ABM) framework as a sociological method, and applied it to church growth with functionalism [24].

We focus on sustainability that correlates with a population change in a religious community in society, specifically, the Presbyterian churches in South Korea. The growth characteristics of a church population basically depend on demographic rates and sustainability factors, such as social economy, political inclination, social capital, environmental emphasis, and concern for future generations.

This article is organized as follows. In section 2, we describe background theories for population dynamics on the basis of the SIR compartmental model in epidemiology. In section 3, we design a prototype model for an SD simulation and explain its key parameters. Additionally, we introduce the ABM framework, analyze key parameters from a micro view, and compare them with those of the SD model. In section 4, we validate the results of population dynamics for an extended SD model with the population trend in the Presbyterian Church of Korea (PCK) as a baseline scenario. We suggest the next scenarios to enhance a church population with two key parameters. In order to impose a sustainability effect into the extended SD model, we adopted 22 indicators in five dimensions that reflect the sustainability of a church. The values of the indicators are obtained by analyzing a number of news articles using a text mining technique. In section 5, we summarize the main insights from this article, and we discuss the role of sustainable potential parameters for the growth of a church population.

## Background theories

According to national survey data on Christianity in the United States and South Korea, the total population with church affiliations is rapidly declining [25, 26]. Most critics diagnose this phenomenon as resulting from internal and external causes. Internally, the decline of the church has occurred because of corruption scandals, embezzlement convictions, sexual abuse allegations, and rape convictions of church leaders. Externally, the churches have been indifferent to social issues, such as conglomerate monopolies, poverty, inequity, and environmental issues on green energy [27, 28].

The 2015 Population and Housing Census in South Korea showed an increase in the Protestant population, but this was not because the church was healthy or evangelized eagerly but due to the cumulative effect of the Protestant population. This is because parents bring their children to church more than other religious groups. In fact, if the cumulative effect is removed, the Protestant population declined by more than 10% [29]. The reason for the decline in the Korean church is due to the political corruption of Christians and the negative social image of the church. Bad incidents against Christianity have made a negative impression on the public and Christianity is recognized as a religion of division among the public.

In a poll on 2018, the 'ethics and practice movement' (45.5%) was cited as the top priority for enhancing the credibility of the Korean church. Volunteer and relief activities recorded the second place with 36.4% [30]. This shows that church contributions and involvement in bettering society help improve the church's image, making people more open to receiving their evangelism efforts.

Thus, the change in the church population reflects how concerned the churches are with social and environmental issues, and how ethical the church leaders are. Analyses of church population dynamics, including sustainability potential, may provide a possible way to achieve

sustainable church growth [31]. In the field of population dynamics, SD and the ABM are useful for exploring population growth and appreciating the implications of key parameters from macro and micro views. In this article, the SD model is developed based on the SIR compartmental model in order to portray the population growth of church members belonging to a specific denomination in South Korea.

## SIR model

The SIR model has been used extensively in the field of epidemiology for analyses of communal diseases, which spread from an infected individual to a population [32]. The epidemic SIR model is composed of three segments: $S$ (susceptible), $I$ (infected), and $R$ (recovered). The SIR model for a closed system assumes that a population is homogeneous, which means each individual makes contact with each of the others randomly. $S$ denotes the susceptible population not yet infected with disease. $I$ represents the infectious population, and it is assumed that infected individuals immediately infect other susceptible individuals, even if they show no symptoms. The contact probability between a susceptible individual and an infected one is proportional to their abundance. $R$ is the recovered population removed from a system by both recovery and death.

These segments of the population are in different stages of the epidemic cycle. The change rate for susceptible individuals is proportional to the product of susceptible individuals and infected ones [33]. In the religious SD model, infectious individuals, $I$, are believers, and susceptible individuals, $S$, are unbelievers. Conversion occurs from contact between believers and unbelievers, just as an infection spreads through contact between individuals in the $S$ and $I$ segments. The recovered individuals, $R$, include church members who left the church or converted to another religion. The total population, $N = S+I+R$, is constant; since a population changes with time $t$, we can express functions of time: $S(t)$, $I(t)$ and $R(t)$. The SIR model forms three differential equations, with $S{\rightarrow}I{\rightarrow}R$ transitions derived by Kermack and McKendrick as follows [34]:

$$\frac{dS}{dt} = -\frac{\beta SI}{N}$$

$$\frac{dI}{dt} = \frac{\beta SI}{N} - \gamma I$$

$$\frac{dR}{dt} = \gamma I$$

where $\beta$ is the transmission rate or the per capita rate of infection, meaning the average number of effective contacts; $\gamma$ is the recovery rate; and the inverse of parameter $\gamma$ is related to the duration of infection, $\tau$. When an epidemic takes place under the condition $dI/dt > 0$, $\gamma/\beta$ is referred to as the threshold of the epidemic. Basic reproduction number $R_0$ is defined as:

$$R_0 = \frac{\beta}{\gamma}$$

## The basic reproduction number

$R_0$ is the number of secondary infections produced by a single infection in a completely susceptible population [35]. $R_0$ determines whether or not an infection might spread through a population and measures the transmission potential of the disease. At the outset of an epidemic, with $R_0 > 1$ and $I(0) > 1$, the infection spreads in the population in the state called

endemic equilibrium. Only when the initial fraction of the susceptible population becomes $S/N > 1/R_0$ can a disease progress. For $R_0 < 1$, the infection dies out over the long term, which is called disease-free equilibrium. The transmission rate, $\beta$, can be expressed as $\beta = cp$, which is the product of two factors: contact rate $c$ and transmissibility of the pathogen, $p$, also referred to as transmission probability [36, 37]. Therefore, the basic reproduction rate is newly defined as $R_0 = cp\tau$. $R_0$ can be reduced by following countermeasures: social distancing, self-isolation, quarantine, and health education programs in order to diminish contact rate $c$; hand washing, face masks, vaccinations, and other hygiene measures against transmissibility $p$; as well as therapeutics, antibiotic treatments for bacterial infections, and boosting the innate immune response for the duration of infectious period $\tau$.

The epidemiological triad (agent, infected host, and environmental factors) provides an inspiration to add parameters related to the availability of public health resources, yet estimation of the $R_0$ value has to be relevant. In a mathematical model, $R_0$ is a survival function for an effective contact rate depending on potential factors across environmental conditions, such as human mobility, contact patterns, social proximity of an infected population, population density, social organization, seasonality, and cultural differences [36, 38]. Community mitigation strategies and political interventions seek to reduce basic reproduction number $R_0$ in order to control a pandemic [39].

In the SIR model for church population dynamics, growth in the number of church members occurs as $R_0$ increases, contrary to the SIR epidemic model. We consider the $R_0$ value to have a sustainable potential, reflecting environmental conditions for the sustainability of a church in society.

## Applications of the SIR model

Hayward investigated a mathematical model for church populations based on the SIR epidemic model, which is useful because an infection process is similar to the process in belief conversion [40]. Church population modeling is categorized into three segments of people: unbelievers, $U$, active believers, $A$, and believers, $B$, with $U \rightarrow A \rightarrow B$ transitions in which church members comprise believers and active believers in a homogeneously mixed system. Active church members evangelize, and recruit unbelievers into the church through the conversion process. A parameter of this conversion potential is related to the basic reproduction potential, $R_0$, in the SIR epidemic model, and increases the number of church members by converting unbelievers. Hayward showed that churches that are growing in the U.S.A. and the United Kingdom have a higher reproduction potential than declining churches [41]. If active believers lose evangelical enthusiasm, they shift to the believers' segment, which is a process called reversion. Hayward considered the effects of reversion, and extended them to the general church-growth model where the population trend has a certain threshold that is associated with sustainability factors [17, 20].

Successful church revivals have occurred when the resistance of the unbeliever community was minimized by effective contact with church members. In Canada, the so-called Toronto Blessing as a church revival had no observable effect on the unbeliever community because of a stigma against churches. In contrast, Alpha Course in the U.K. had a nationwide revival impact, then spread to 169 countries around the world. Alpha Course focused on practices inviting unbelievers to hospitable small meetings, and endeavored to improve the social environment through radio and TV broadcasting. It was an example of how a social effect overcame the threshold for a revival of the church through the media [20].

Ochoche and Gweryina [22] and Madubueze and Nwaokolo [21] also used the SIR epidemic model for church population dynamics, including susceptible unbelievers $S$, passive

believers *P*, and active believers *A*, all with an $S \rightarrow P \rightarrow A$ transition model. The susceptible unbelievers included non-church members and church members from other denominations, and they uniformly contacted church members. Ochoche and Gweryina [22] showed that two key parameters contribute to an increase in a church population: the effectiveness of active believers, and the rate of infectious enthusiasm. In this model, the effectiveness of active believers is related to the role of the reproduction potential in the SIR epidemic model. Active believers are mature church members who have completed a discipleship course and are more dedicated than passive believers.

## Methodology

System Dynamics combines mathematics and computer simulation to explore behavior in real-world systems, relationships, and processes over time. The SD approach is an effective method to formalize a system structure, and it provides a better understanding of what drives behavior in a system. In addition, it can examine future dynamics based on a given set of assumptions [42]. The SD model introduced by Forrester is an operative method to reveal behavior in a complex system, considering a multi-loop and feedback structure with non-linearity [43]; the SD model is useful for addressing population dynamics by characterizing model construction, analyzing the mathematical model, and forecasting a scenario performance that identifies critical feedback influencing the system [44].

### Model construction

We designed a population growth model reflecting the current church structures in South Korea, which are operated by two wings of small groups and massive worship meetings. The so-called House Church has a positive influence on Korean churches by concentrating on evangelism toward unbelievers, and nurturing passive believers to become active believers who serve the church as small-group leaders [16].

Fig 1 illustrates a stock-flow prototype diagram for a church population. The SD prototype model includes three stocks (unbelievers *U*, passive believers *P*, and active believers *A*), and four flows (loss of belief, evangelism, conversion, and discipleship) with variables connected by arrows. Total population $N = U(t)+P(t)+A(t)$ is constant. Evangelism flow occurs between unbelievers and passive believers, and conversion flow acts between unbelievers and active believers. The prototype model is reduced to differential equations as follows:

$$\frac{dU}{dt} = \mu A - \frac{sPU}{\tau_e N} - \frac{sAU}{\tau_c N}$$

$$\frac{dP}{dt} = \frac{f_e sPU}{\tau_e N} - \delta P$$

$$\frac{dA}{dt} = \delta P + \frac{f_c sAU}{\tau_c N} - \mu A$$

with $f_e$ = evangelism fraction, $f_e$ = conversion fraction, $\tau_c$ = conversion duration, $\tau_e$ = evangelism duration, $\mu$ = loss rate, $\delta$ = disciple rate, and $s$ = sustainable potential.

In the present SD model, new church members are generated through evangelism and conversion of unbelievers. We define evangelism as taking place when passive believers contact unbelievers in a society with evangelism fraction $f_e$, and conversion occurs when active believers create inner change in unbelievers with conversion fraction $f_c$. Hayward fixed the following sum: $f_e + f_c = 1$ [17].

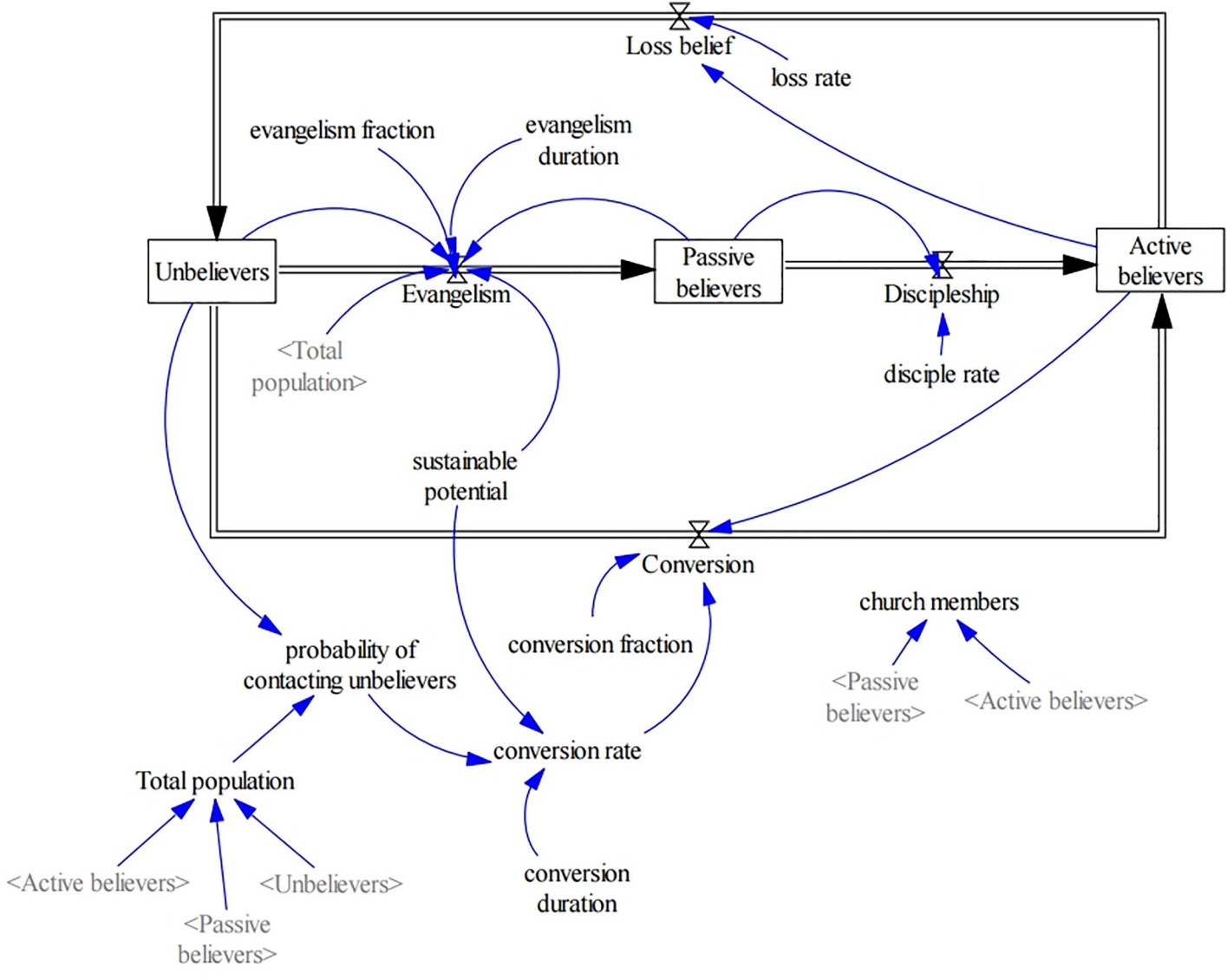

**Fig 1. Stock and flow prototype diagram for a church population (S1 Fig).**

On the other hand, active believers who lose enthusiasm become unbelievers at loss rate $\mu$ through the loss-of-belief flow. Evangelism duration $\tau_e$ is a time interval in which passive believers evangelize unbelievers, and similarly, conversion duration $\tau_c$ is the time interval in which active believers convert unbelievers. Sustainable potential $s$ was called conversion potential by Hayward [17], and was called effectiveness of active members by Ochoche and Gweryina [22]. Sustainable potential $s$ reflects how amicable unbelievers are to the Christian faith, and integrates social factors in a society. When $s$ is low, church members have very little effective contact with unbelievers, and the tendency to evangelize is weak.

## Model analysis in a system dynamics framework

A model simulation was implemented with Vensim software, developed by Ventana Systems, Inc. The SD prototype model had a total population of 1,000 and was simulated over a 30-year

time period with intervals of 0.125 years. The initial church members, including passive believers and active ones, was 5% of the total population. The initial number of active believers had no influence on the threshold for rapid population growth. The threshold only depended on sustainable potential $s$, which increased the number of effective contacts with unbelievers. An increasing sustainable potential lowers the threshold, then accomplishes rapid growth in the church population in the early stages [20].

Fig 2 shows the variations in the populations of passive believers and church members when sustainable potential $s = 1, 2,$ and $3$, and where other parameters were fixed as follows: $f_e = 0.5, f_c = 0.5, \tau_c = 5, \tau_e = 2, \delta = 0.25,$ and $\mu = 0.01$. The threshold of the church population is obtained when $\frac{dP}{dt} > 0$, that is, $s > (\tau_e \delta / f_e)(N/U_0)$, where $U_0$ indicates the initial number of unbelievers. The threshold condition was $s > 1.31$ for the initial parameters. When $s$ is smaller than the threshold, effective contact between unbelievers and church members decreases. The population growth faces a long lag phase, declining asymptotically for $s = 1$, as shown by the black dotted line in Fig 2. When $s$ equals 2 and 3, passive believers contribute to rapid population growth, seen as the red line for $s = 2$ and as the green line for $s = 3$.

When the initial population of unbelievers, $U_0$, exceeds the threshold determined by sustainable potential $s$, the rapid growth of active believers shows an S-shaped curve, and the population of passive believers is the bell-shaped curve. A church population in the epidemic cycle underlies the stabilizing processes as a limit, where the S-shape curve with exponential growth occurs, and finally accomplishes the asymptotic church population at the final equilibrium state.

## Analysis in the agent-based model framework

The System Dynamics model is basically deterministic, and is mainly used at the strategic or conceptual level. The SD model treats simulated objects as a continuous mass, and the method is designed with top-down feedback to be highly simplified and abstract. The SD model for church population dynamics can account for church growth, yet overlooks complex individual mechanisms and the trajectories associated with conversion behavior.

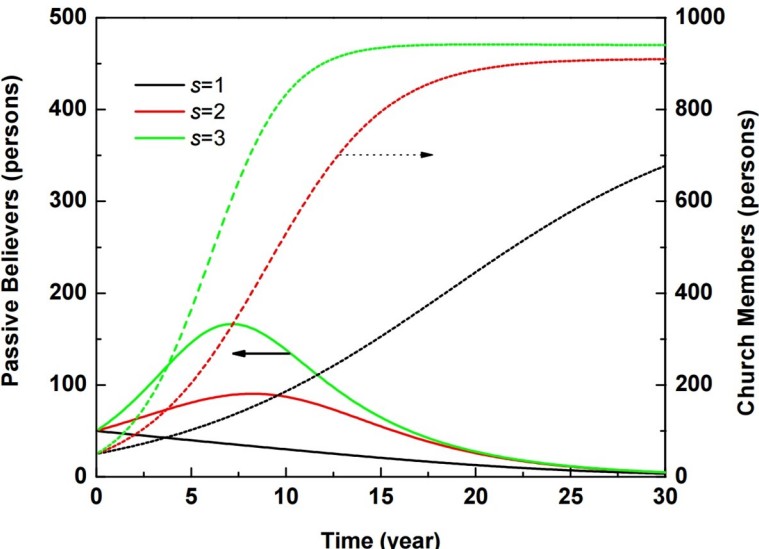

**Fig 2. Variations in passive believers and church members based on $s = 1, 2,$ and $3$. Solid lines depict passive believers, and the dotted lines denote church members (S2 Fig).**

Unlike the SD model, the agent-based model is a bottom-up computational method. The ABM uses an individual entity represented by discrete agents in a simulated space, where agents autonomously produce emergent and non-intuitive outcomes at the population level. Interactions with agents are set in predefined rule-based algorithms at the model development stage. The ABM may be more appropriate for explaining micro behaviors in a system [45], and is a useful way to understand the social systems that grew out of complexity science and artificial intelligence [46]. The ABM is an approach to overcome the limits of the SD approach, especially in the case of a system containing active objects, such as people, animals, and vehicles, with timing or other kinds of individual behaviors [47].

We used the ABM framework already developed by the open-source NetLogo software to simulate the spread of HIV/AIDS, which concentrated only on the infection of HIV via sexual contact, generating population progressions in one-week cycles [48]. The SIR approach well explains the transmission of HIV infection [49]. The simple HIV ABM framework limits the consideration of only typical agent contact without contact patterns in the real world, such as contact between family members, classmates, and co-workers [50]. The simple ABM framework does not include 'data culture' to generate agent contact and movement patterns like intelligent transportation system, cellular service provider data, personal information on social media and technologies that may leverage smartphone and another mobile device [51].

We customized the model to fit church population dynamics, and modulated the global variables of average couple commitment, which means the duration when staying in an infected agent and susceptible agent couplehood. The average couple tendency indicates how much one agent likes becoming a couple, and then, the couples form and break up according to coupling tendency and couple commitment. The likelihood of agent coupling determines the degree of transmission [52].

In the initial conditions of the ABM framework, 1,000 church members were set as the total number of agents, with 5% of the church members passive believers for over 30 years, similar to the conditions in the SD prototype model. Church members are composed of passive believers and active believers, where the former group compares to agents who are infected but unaware of the infection, and the latter group includes agents who are infected and aware of the infection. Couple commitment was fixed at one year, and the duration of symptoms that show up after infection was four years, which is related to disciple rate $\delta$ in the SD prototype model.

Agents as church members wander over the modeled space when they are not in couples. When coming into contact with a susceptible unbeliever, there is a chance the two agents will couple, which simply means there is effective contact between the church member and the unbeliever. The church population is proportional to the number of effective contacts, just as the spread of an infectious disease is proportional to the number of contacts with a coupling tendency and couple commitment. We realize that the sustainable potential parameter in the SD prototype model is related to the coupling tendency in the ABM framework. Fig 3 shows that the population of passive believers declines asymptotically for $s = 1$, as depicted by the black thick line, where the decreasing passive believers do not contribute to rapid church growth. It means the threshold of rapid population growth exists when $s \in [1,2]$, like that of the SD prototype model.

The ABM-approaching church population goes through an S-shaped curve with exponential growth and reaches stabilization in 25 years for $s = 2$ and 17 years for $s = 3$ as shown in Fig 3. This is in good agreement with the results of the SD approach as shown in Fig 2. The exponential growth of the S-shaped curve means a great revival of church. This result indicates that the sustainable potential $s$ must be above the threshold in order to achieve continuous stabilization after the great revival inside one generation.

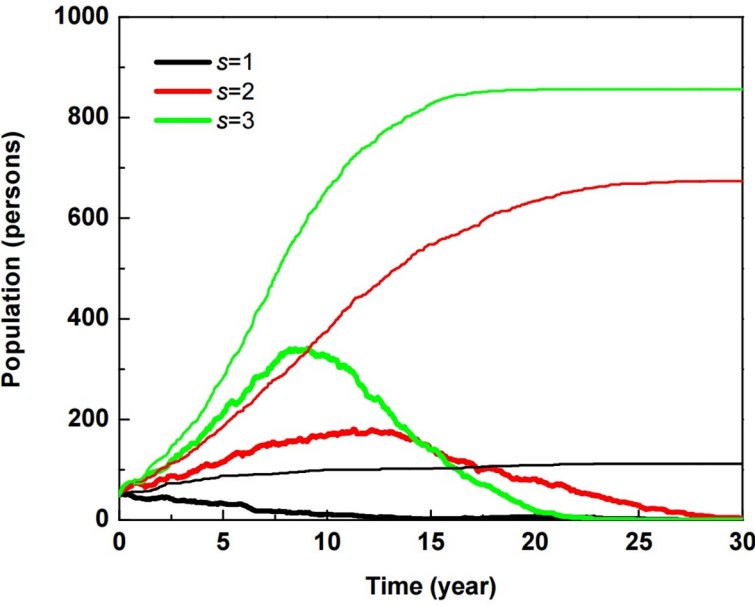

**Fig 3. Population of church members (thin line) and passive believers (thick line) for disciple rate $\delta$ = 0.25, 0.33, and 0.5 in the ABM framework with sustainable potential $s$ = 1 (S3 Fig).**

In the church population SD model on SIR approach, it is not easy to understand what the parameter changes over specific time actually mean. We investigated the role of the disciple rate for church members in the ABM framework, because the ABM approach is useful for understanding the effect of time changes. Ochoche and Gweryina showed the disciple rate contributes to increasing the number of active believers [22]. Disciple rate $\delta$ is defined as the reciprocal of the discipleship period, which corresponds to the period an infected individual can start to show symptoms after infection in the human immunodeficiency virus (HIV) ABM model. The ABM framework on the SIR approach provides a micro perspective to understand changes in the number of contacts over specific time changes. In the discipleship process, passive believers can mature into active believers, where the process usually lasts four years in church situations in South Korea [16]. When the duration varies among 200, 150, and 100 weeks in the ABM framework, disciple rate $\delta$ equals 0.25, 0.33, and 0.5, respectively. Church members and passive believers decrease with the increasing disciple rates, although the disciple rate contributes to an increase of active believers. This result implies that too many active believers may hinder church growth. In fact, the majority of the congregations in the House Church have a small group of leaders at about 10–20% of active believers, because too many leaders in a small group can be counterproductive [16].

## Results

### Application to the Presbyterian Church of Korea

The Christian population in South Korea is 20.3% of the total population, according to the 2017 national census [53]. The Presbyterian Church of Korea is the largest denomination of Christian churches in South Korea. The number of Christians in South Korea started to decrease at -1.4% starting in 2005, and PCK members declined after 2009 as well. Some PCK churches became inactive and did not recruit enough new members to sustain themselves [28, 54]. In order to stabilize and avoid extinction, the segment of believers needed to be

replenished, which can happen when those who left the church rejoin it, and when new-born children become Christians [55]. Church growth is not only related to the size of the church but also to its maturity. In early Korean church history, churches experienced a rapid growth in the so-called Great Revival. The churches built hospitals to cure the sick, educated elementary school children, and spread agricultural technologies to solve food problems. Furthermore, they established welfare organizations for the aged, set up orphanages, and started daycare centers. But now, the social effectiveness of the church in each local area is again at an elementary level. One of the cues for church growth in South Korea is when the churches organize welfare activities to fix social problems, and assist in addressing common needs in the community [55].

**The extended church-population SD model.** The SD prototype model can be extended to apply to real church population resources, as shown in Fig 4. Bidirectional flow between unbelievers and passive believers indicates that passive believers evangelize unbelievers, and in the opposite direction, passive believers revert to unbelievers. Likewise, bidirectional flow between unbelievers and active believers means active believers can be inactivated to become unbelievers, and unbelievers can be converted into active believers.

The generic SD model for a church population includes demographic factors, such as birth rate, $b$, and mortality (or death rate), $m$, the stock of other religious people, $O$, the reversion flow, and the inactive flow. For mortality, the simple church population SD model assumes that birth rate and mortality rate can be ignored when considering short-term church growth of about 15 years. However, the extended models that include long-term effects over a generation should take into account birth rates and mortality rates [17]. Church members are assumed to hold onto their belief for their whole lifetime [40]. *Other religious people* mean

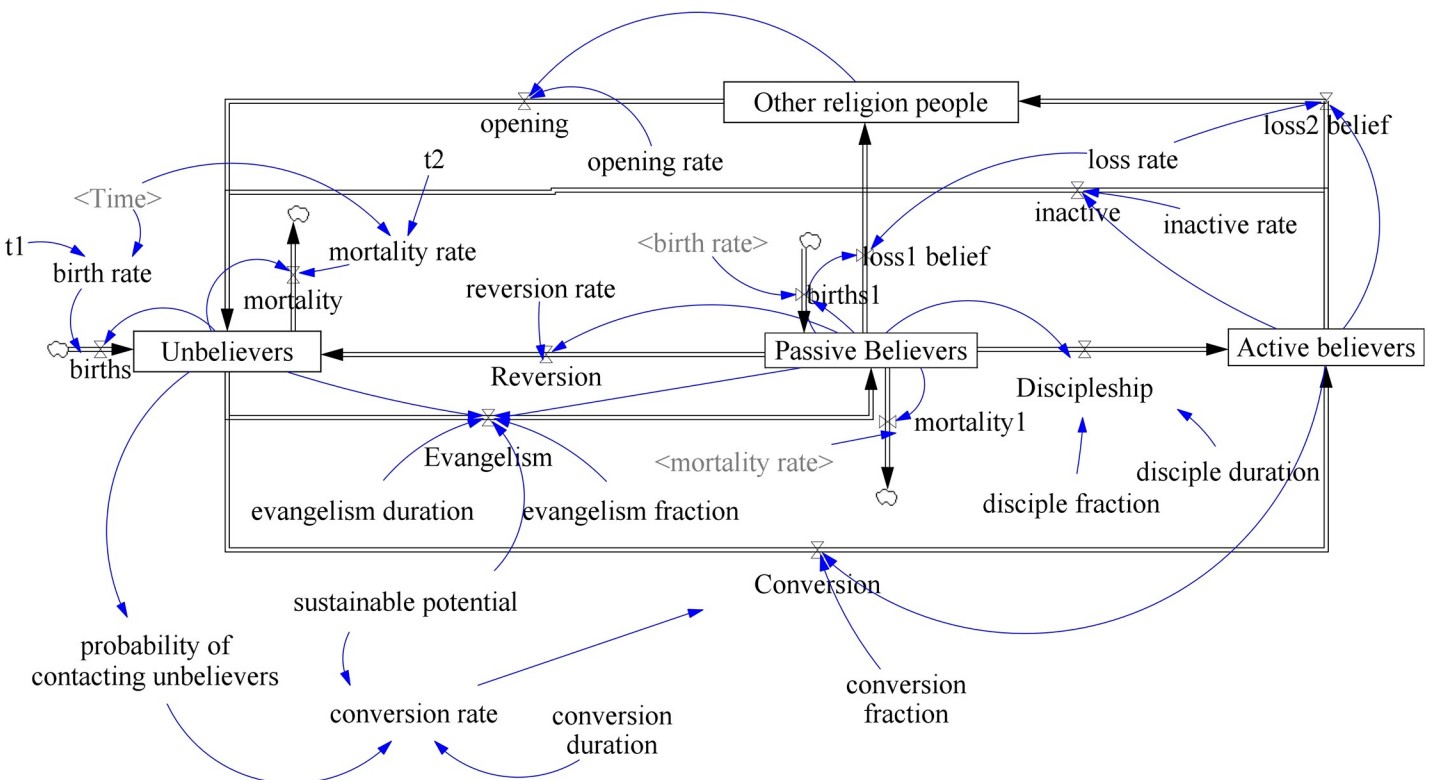

**Fig 4. Extended stock and flow diagram for a church population extended to apply to real church population resources (S4 Fig).**

people who are not PCK church members, and includes other Christian denominations, such as Baptists, Full Gospel, Methodists, etc. Some passive believers shift to active believers through the discipleship flow, and others revert to unbelievers at the reversion rate, $\eta$, or leave the church and convert to other religions at the loss rate, $\mu$. Likewise, active believers become inactive at the inactivity rate, $\omega$, to become passive believers, or they become unbelievers after losing faith, at loss rate $\mu$. The loss of belief flows reflects how the rest of the active believers shift to other denominations and religions, although the rates are negligible [56]. Some other religious people who open their hearts to Christianity become susceptible unbelievers at opening rate $\rho$ through the opening flow. Similar to the stock of other religious people, Madubueze and Nwaokolo added inactive Christians stock $I$ to examine the effects of reversion and renewal on the church population in the mathematical $U{-}P{-}A{-}I$ model that included birth and mortality rates [21], and Hayward labeled this kind of people *hardened* [20].

The differential equations of the generic SD model for church populations are extended as follows:

$$\frac{dU}{dt} = \omega A + \rho O + bU - mU + \eta P - \frac{sPU}{\tau_e N} - \frac{sAU}{\tau_c N}$$

$$\frac{dP}{dt} = bP - mP + \frac{f_e sPU}{\tau_e N} - \frac{f_d}{\tau_d}P - \eta P - \mu P$$

$$\frac{dA}{dt} = \frac{f_d}{\tau_d}P + \frac{f_c sAU}{\tau_c N} - \mu A - \omega A$$

$$\frac{dO}{dt} = \mu P + \mu A - \rho O$$

where $N = U{+}P{+}A{+}O$ is the total population (with a function for time), and disciple rate $\delta$ is decomposed into disciple fraction $f_d$ and disciple duration $\tau_d$. Population dynamics generates the total church members of the PCK for a baseline scenario.

**Model parameters analysis.** Initial data on church stocks were obtained mostly from reference to PCK annual reports [26] and KSIS reports [53]. In 1995, South Korea's population was 44,553,710. The religious population included Buddhists, Catholic and other Christian believers, and Confucianists at 22,597,824. Initial passive believers who were church members of the PCK in 1995 totaled 2,105,000. Initial active believers comprised 5% of the initial passive believer population of the PCK (105,000). The term *church members* represents the sum of passive believers and active believers. Initial parameters of flows and other auxiliary variables are optimized to reproduce the trend of the PCK population from 1995 to 2018. We conducted the optimization test called as the one-at-a-time method to fix the values of other variables when optimizing the value of one parameter [57]. The optimization test is basically allowing to adjust the parameters involved in the model so it produces a minimal error according to some error metric on a specific test dataset. We determine the set of parameters by comparing simulation results with reference data, so that the difference between simulation output and reference data is as small as possible [58].

The model validation was performed comparing the model's results with the historical data. A reference for the validation-process of the church population growth SD model is obtained by the total population data of the Presbyterian Church of Kore (PCK) for 1995–2018. We compared the PCK data with the simulation results of the SD model for validation. The Mean Absolute Percentage Error (MAPE) metric is used to validate the SD model and the threshold

**Table 1. Parameters and values of model.**

| Parameter | Baseline scenario value | Scenario 1 value | Scenario 2 value |
|---|---|---|---|
| Inactive rate ($\omega$) = loss rate ($\mu$) = opening rate ($\rho$) = 0.01 | | | |
| Sustainable potential ($s$)* | 1 | 1.2 | 1.3 |
| Reversion rate ($\eta$) | 0.05 | 0.05 | 0.05 |
| Disciple fraction ($f_d$) | 0.16 | 0.16 | 0.16 |
| Evangelism fraction ($f_e$) | 0.22 | 0.22 | 0.22 |
| Conversion fraction ($f_c$) | 0.01 | 0.05/0.1 | 0.05/0.1 |
| Disciple duration ($\tau_d$) | 3 | 3 | 3 |
| Evangelism duration ($\tau_e$) | 1 | 1 | 1 |
| Conversion duration ($\tau_c$) | 5 | 5 | 5 |

*For the Baseline Scenario at $s = 1$, for Scenario 2 at $s = 1.2$, and for Scenario 3 at $s = 1.3$.

for acceptable MAPE values is 10%. The MAPE is represented by following equation;

$$MAPE = \frac{1}{N} \sum_{t=1}^{N} \frac{|A_t - F_t|}{A_t} \times 100$$

where $A_t$ is the actual value and $F_t$ is the forecasted value at time $t$, and N is number of periods. For the baseline scenario for $s = 1$ (and the referenced data) $MAPE = 4.4\%$ and an $R^2$ value of 0.85 are observed during validation, which implies reasonable measure for the SD model [59].

The values of the initial parameters were calibrated to fit the PCK population from 1995 to 2018. In order to prevent overfull parameters, inactive, loss, and opening rates were assumed to retain 1% [17]. Based on the baseline scenario values listed in Table 1, we analyzed what key parameters contribute to the growth of a church population. Fig 5 shows the historical population of the PCK (bar graph) [26]. Real birth rate (solid squares) refers to the crude birth rate

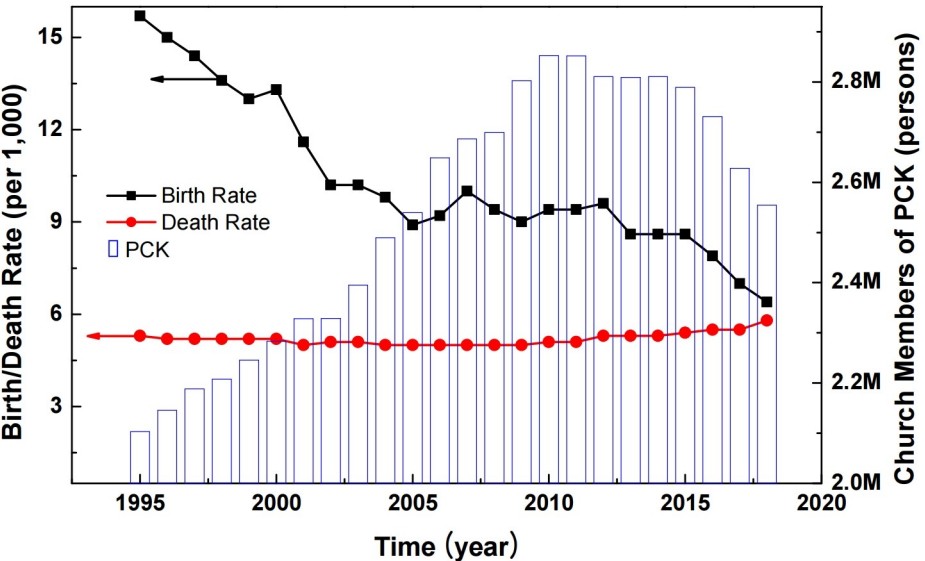

**Fig 5. The real birth and death rates for South Korea between 1995 and 2018 are depicted by solid squares and solid circles, respectively.** The historical population of the PCK is illustrated with the bar graph (S5 Fig).

(CBR), and real death rate (solid circles) indicates the crude death rate (CDR); both are measured against the birth and death rates per thousand persons [53].

**Measurement of sustainable potential.**   Churches on a national scale are complex systems consisting of social, institutional, political, economic, geographic, educational, cultural, environmental, and ecological elements interacting in dynamic ways [60]. These elements are related to sustainability factors in a society, which can contribute to enhanced church populations. One of the primary functions of a church is to help people deal with adversity and to serve society by providing necessities [61]. The external considerations of the church as a social force markedly influence the growth of the church population [17]. The direct and indirect contributions to sustainability in a church increase its potential social impact and bestow satisfaction on church members. The potential social impacts can increase a favorable impression of the church for unbelievers. The positive influence for sustainability produces a bilateral attraction between unbelievers and church members, then works based on the conversion potential identified by Hayward [20], based on the effectiveness of active believers identified by Ochoche and Gweryina [22], or like the couple tendency in the ABM framework. These parameters represent the basic reproduction rate in the SIR epidemic model. In this article, we define the potential social impact as the parameter of sustainable potential.

We determined the value of the sustainable potential as a sustainability measurement for the church in South Korean society. Although the sustainable potential in the previous SD model for a church population is fixed as a constant over all years, we identify a time-dependent sustainable potential depending on the year. Sustainability measurement poses a number of methodological difficulties and limitations related to the selection of indicators, to data processing, and the interpretation of results [9]. Many diverse indicators are adequately characterized in the sustainability pillars of the environment, the economy, and society, which are interrelated and time-dependent [62]. There is no standard for the choice of indicators increasing in the direction needed most. Additional indicators can be selected to better reflect quality of life. We investigated suitable and efficient indicators of sustainability for a church, based on the principles of relevance, comprehensibility, reliability, and availability, as suggested by Lior et al. [9].

In the present article, the sustainability indicators of a church were selected from scholarly works related to practices for sustainability [5, 9, 63–66]. We decided on the 22 representative indicators in the five sustainability dimensions consisting of economic, educational, social, political and environmental domains as defined in Fig 6. Sustainability indicators refer to social environmental factors and elements within the church that directly and indirectly affect the sustainability of church.

Among sustainability indicators, the church's financial income is collected by voluntary donation, bazaar fundraising, and personal fundraising or offering. Church education takes place individually through counseling and collectively through Sunday School education, and the expenditure of scholarship in church finance plays a role in promoting church education by supporting seminary students. The church's social activities are carried out by volunteers who deliver relief supplies to the elderly, the disabled, the poor, single parents, and the victims of the disaster. The political activities of church are conducted by participation in public hearings and elections to realize social justice. The church's interests in the field of environment come from the creation faith and appear in all environmental movements aimed at preserving the nature that God created.

An increased level of media coverage in newspapers on sustainability-related issues may indicate that an increased level of public awareness contributes to the issues [67, 68]. Newspaper articles cover a number of different aspects, including general values about a society, and deal with sustainability, such as social and political issues that are regarded as newsworthy by journalists and press agencies [69]. Although the quantitative information from news articles

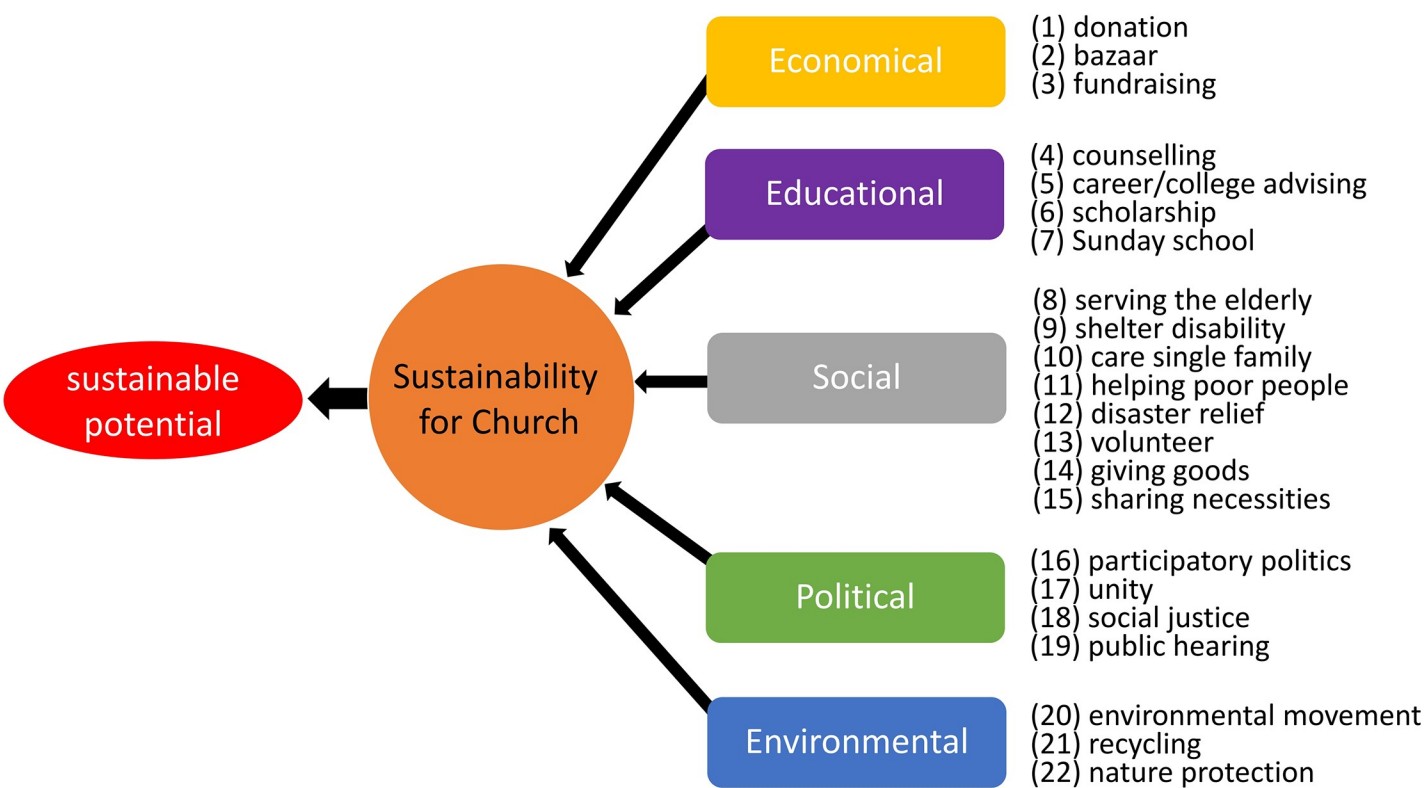

**Fig 6. Schematic for obtaining the sustainable potential from church-related sustainability indicators (S6 Fig).**

has to be treated with caution, the frequency of key terms in news articles for the different factors plays an important role in comprehending the meanings and functions of sustainability.

We selected 11 national newspapers and 28 regional newspapers written in the Korean language, and performed keyword searches for indicators related to pivotal practices in the field of sustainability. All articles contained at least one mention of a search keyword within the body text. We measured the number of news articles related to churches for consecutive years from 1995 to 2018, where data were collected by text mining techniques based on morphological analysis [70].

The total number of news articles is searched by keyword *church* but not *Catholic Church* between 1995 to 2018. News articles were searched for keywords representing indicators for sustainability of a church from the BIGKinds operated by the Korea Press Foundation [71]. Values of the indicators were obtained from analyses of a number of newspaper articles published between 1995 and 2018 that included the indicators as keywords (S1 Table).

The total number of newspaper articles searched from 1995 to 2018 using 22 sustainability indicators as keywords in BIGKinds system. In order to apply news big data, which is an unstructured text, as analysis data, the data must be processed so that Natural Language Processing (NLP) can be performed. For data cleaning, the authors perform morphological analysis using machine learning and morpheme analysis dictionaries, especially using noun dictionaries built inside the BIGKinds system. The BIGKinds system uses a structured Support Vector Machine (SVM) algorithm for morphological analysis and data preprocessing. Automatically extract all noun keywords from the article text and remove stopwords. The structured SVM algorithm is a machine learning algorithm for NLP of text and shows 97.13% performance in Korean predicate recognition and classification.

In the present study, the values of the indicators represent positive sustainability attributes. We adapted the sustainability measurement method applied by Lior et al. which normalizes each indicator value, $z$, by using the min-max method for the raw indicator values, $x$, as follows [9]:

$$z_j = \frac{x_j - x_{min}}{x_{max} - x_{min}}$$

Normalization was implemented for each indicator type $j$ with $j = 1$ to $J$, where $J$ represents 22 indicators of sustainability; $z_j$ values are the normalized indicators, and are positive by definition; $x_j$ values are raw, or pre-normalized, indicators; $x_{min}$ is minimal value; and $x_{max}$ is the maximum indicator of the raw indicators. In order to transplant the normalized values of the indicators into the SD model for a church population, the average values of time-dependent sustainable potential $s_t$ were scaled by $s_{max}$, which was obtained when the simulated number of church members reached equilibrium in the SD model simulation, $s_{max} = 4$, and the equation reduces to

$$s^t = s_{max} \cdot \frac{1}{A^t} \sum_{j=1}^{J} z_j^t$$

where $s^t$ is the time-dependent sustainable potential, $z_j^t$ is the normalized indicator value, and $A^t$ is the number of total articles searched for *church* at time $t$.

Fig 7(A) shows the time-dependent sustainable potential from 1995 to 2018. We compared the church members simulated by using the time-dependent sustainable potential to the historical population of the RCK, as shown in Fig 7(B). The values for sustainable potential after 2018 were estimated based on a linear increase of up to 1.5 times the value in 2018, and other parameters were slightly optimized. Results show the modified church population (solid red circles) seem to reproduce the specific characteristics of time for the PCK population (solid black squares). We can see that the average values of the time-dependent sustainable potential obtained by analysis based on the number of news articles may reflect the level of sustainability.

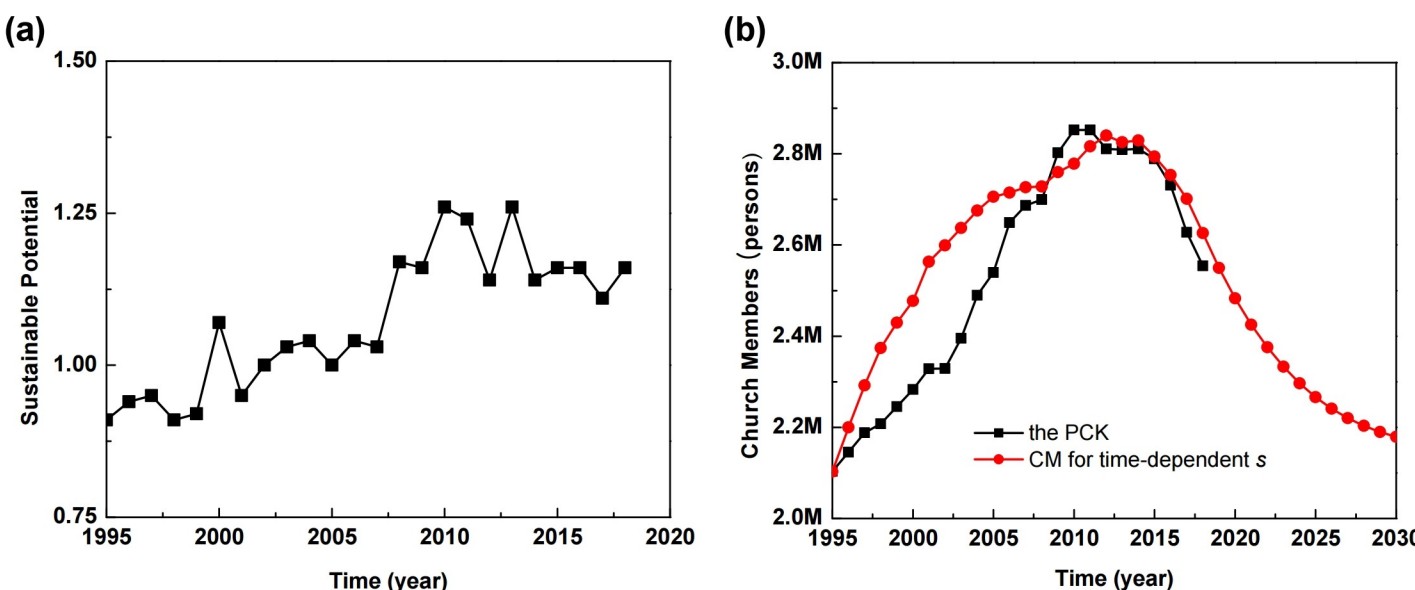

**Fig 7.** (a) The time-dependent sustainable potential from sustainability measurements (S7 Fig). (b) Simulated church members for time-dependent sustainable potential (red solid circles), and the church population of the PCK (solid squares) (S8 Fig).

In addition, as a key parameter for church growth, we investigated an effect of conversion fraction $f_c$ that indicates unbelievers are converted by active believers, which discretely increased from 0.05 to 0.4. We found there are two key parameters for sustainable potential and conversion fractions, where the former contributes to a dramatic increase in church members, and the latter causes minor growth, similar to Hayward's results [20]. We can define sustainable potential, corresponding to basic reproduction number $R_0$ in the SIR epidemic model, as an external factor for church growth, because it relates to the social environment and policies, whereas the conversion fraction becomes an internal factor reflecting effective contacts by active believes in converting unbelievers to church members.

We considered scenarios manipulated by the two key parameters for sustainable church growth in the near future. Fig 8 shows simulated results (black solid line) compared to the historical trends in PCK church members (solid squares) which declined from 2010 and did not reach equilibrium. Scenarios 1 and 2 designate mixed strategies focusing on two key parameters to enhance church populations with other optimized parameters listed in Table 1. Scenario 1 is the case where conversion fraction $f_c$ has the values of 0.05 and 0.1 for a sustainable potential of $s = 1.2$. As shown in Fig 8, for scenario 1, the simulated populations with $f_c = 0.05$ and $f_c = 0.1$ enhance the population to 16% and 20% in 2030, compared to the baseline scenario, respectively. Likewise, in scenario 2, the simulated populations with $f_c = 0.05$ and $f_c = 0.1$ for $s = 1.3$ lead to increases of 21% and 25% in 2030, compared to the baseline scenario. The results prove that the dual action of increasing the sustainable potential and the conversion fraction play a key role in the growth of a church population in near future.

## Conclusions

The results for church population dynamics presented in this article for the Presbyterian Church of Korea provide insight into, and comprehension of, the relationship between religious affiliation and sustainability. Simulations based on the Sustainable–Infected–Recovered

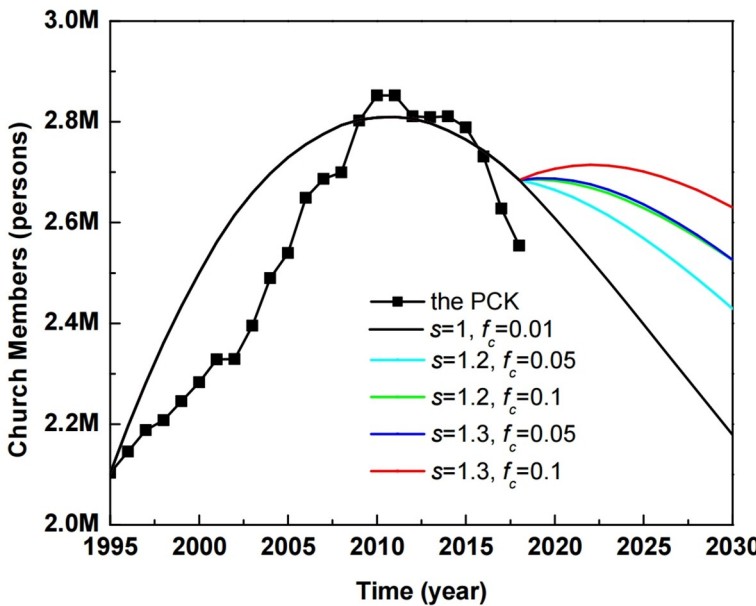

**Fig 8. Baseline scenario (solid line) with $s = 1$ and $f_c = 0.01$ compared to the historical population of the PCK from 1995 to 2018 (solid squares).** Scenario 1 for $s = 1.2$ is shown with cyan and green solid lines, when $f_c = 0.05$ and 0.1, respectively. Scenario 2 for $s = 1.3$ appears with blue and red solid lines when $f_c = 0.05$ and 0.1, respectively (S9 Fig).

epidemic model are carried out in the System Dynamics and ABM frameworks, which offer a micro-view analysis of the key parameters, and reinforce an understanding of the role of a parameter. An increase in coupling tendency in the HIV/AIDS ABM framework is comparable to the conversion potential of active believers in the SD model for a church population, where the values for coupling tendency and conversion potential represent the basic reproduction number in the epidemic SIR model. Sustainable potential in the present SD model reflects the role of the reproduction number, including potential social factors that influence the contact rate, such as socio-behavioral and environmental circumstances [72].

According to the results, there are two key parameters of sustainable potential as an external effect, and of the conversion fraction as an internal effect, to enhance a church population. The sustainable potential leads to dramatic growth of a church population, and the conversion fraction contributes to minor growth. The comprehensive growth of a church population occurs when unbelievers continue to become church members through enthusiastic contact with active believers, just as individuals susceptible to disease are infected by contact with infectious individuals. The increasing conversion fraction related to active believers means that active believers as small-group leaders are more committed to evangelism and conversion of unbelievers, instead of remaining in a decision-making group for church governance and/or volunteering in other departments.

High sustainable potential exerts a positive social influence, and increases effective contacts with unbelievers, and then contributes to dramatic growth of a church population, which indicates that sustainable potential works as an external factor of church growth. There is a complex relationship between religious affiliations and socio-demographics from the aspect of sustainability concerning time. Yet, previous SD model research on church populations neglected the time effect of sustainable potential. We take into account the time-dependent sustainable potential through the sustainability measurement, which numerates the values of indicators for a level of sustainability. We inferred the values of indicators for sustainability by analyzing a number of newspaper articles based on keyword search through a text-mining approach. Normalized values for the indicators were scaled and substituted into the time-dependent sustainable potential in the SD model. Church populations with time-dependent sustainable potential specifically reproduced the characteristics of time in the PCK church population. We can conclude that sustainability acts as a positive factor in the growth of church populations based on the SD model simulation. This result also suggests that the decision-makers of the church should plan external activities to contribute to the sustainability of society in order to maintain sustainable church growth.

## Limitations and future works

In this paper, the function of sustainability is represented by the sustainable parameter. Raising the sustainable parameter is a way to achieve the sustainability of the church population because the church population stabilizes when the sustainable parameter exceeds the threshold. However, the discussion on how to increase the sustainable parameter by adjusting 22 sustainability indicators goes beyond the scope of this paper, as it is expected to undergo complex processes requiring big data analysis.

We produce the time-dependent sustainable potential as external factors using text mining techniques for newspaper articles over years. However, the value is inappropriate to monitor changes over a short period of time. The prompt information extracted from social media communication is suitable for predicting the time-dependent sustainable potential influencing the sustainability of church population due to the short-term change such as the COVID-19 pandemic.

Social media-based relationships can lead to positive or negative word of mouth (WOM) recommendations. Social media interactions between believers and non-believers affect how non-believers think and feel about the church, and consequently these interactions can lead to the positive WOM, affecting the conversion of non-believers. We can find and measure external factors based on responses on social media such as Facebook affecting the sustainability of the church population [73–76]. In future study, we can discuss the sustainability of the church population on the COVID-19 SIR SD framework using the time-dependent sustainable potential obtained from social media communication.

## Supporting information

**S1 Fig. Document for prototype SD model.**
(DOCX)

**S2 Fig. Document for church members on SD.**
(DOCX)

**S3 Fig. Document for church members on ABM.**
(DOCX)

**S4 Fig. Document for extended SD model.**
(DOCX)

**S5 Fig. Document for real birth rates and death rates.**
(DOCX)

**S6 Fig. Document for schematic with sustainable potential.**
(PPTX)

**S7 Fig. Document for time-dependent sustainable potential.**
(DOCX)

**S8 Fig. Document for simulated church members.**
(DOCX)

**S9 Fig. Document for church population for scenarios.**
(DOCX)

**S1 Table. Document for the number of news articles.**
(DOCX)

## Author Contributions

**Conceptualization:** Chulsu Jo, Doo Hwan Kim.

**Data curation:** Chulsu Jo.

**Formal analysis:** Chulsu Jo.

**Funding acquisition:** Doo Hwan Kim, Jae Woo Lee.

**Investigation:** Chulsu Jo.

**Methodology:** Chulsu Jo, Doo Hwan Kim.

**Resources:** Chulsu Jo, Jae Woo Lee.

**Software:** Chulsu Jo.

**Supervision:** Jae Woo Lee.

**Validation:** Chulsu Jo.

**Visualization:** Chulsu Jo.

**Writing – original draft:** Chulsu Jo, Jae Woo Lee.

**Writing – review & editing:** Chulsu Jo, Doo Hwan Kim, Jae Woo Lee.

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
