## [Decision Letter · Decision Letter 0]

1 Jan 2021

PONE-D-20-30060

Church population dynamics with sustainability

PLOS ONE

Dear Dr. Lee,

Thank you for submitting your manuscript to PLOS ONE. After careful consideration, we feel that it has merit but does not fully meet PLOS ONE’s publication criteria as it currently stands. Therefore, we invite you to submit a revised version of the manuscript that addresses the points raised during the review process.

Both reviewers liked the use of mathematical modeling to address a social issue like church membership, but they also feel that the manuscript in its present form suffers from many flaws. In preparing your revision to address all comments by the two reviewers, I hope you would give special attention to: (1) your choice of title for the manuscript, since its scope is significantly narrower than the question on sustainability of churches; (2) the framing of your study, as both reviewers feel that the present perspective is rather negative, but at the same time, there is tremendous potential to use the mathematical modeling more positively. I understand that this rethinking about the manuscript would take time. Please do not hesitate to ask for more time if you feel you need it.

We look forward to receiving your revised manuscript.

Kind regards,

Siew Ann Cheong, Ph.D.

Academic Editor

PLOS ONE

Journal Requirements:

Reviewers' comments:

Reviewer's Responses to Questions

**Comments to the Author**

1. Is the manuscript technically sound, and do the data support the conclusions?

Reviewer #1: Partly

Reviewer #2: Partly

2. Has the statistical analysis been performed appropriately and rigorously? 

Reviewer #1: N/A

Reviewer #2: I Don't Know

3. Have the authors made all data underlying the findings in their manuscript fully available?

Reviewer #1: No

Reviewer #2: Yes

4. Is the manuscript presented in an intelligible fashion and written in standard English?

Reviewer #1: Yes

Reviewer #2: Yes

5. Review Comments to the Author

Reviewer #1: Overall comment

We are very sorry to have to reject it this time, but we think it would be a tremendous improvement if you could make better use of the Korean original data set and present a data set that can be used for empirical experiments.

We also agree with the theme of the paper and the ideology of the abstract.

However, in agreeing with you, we find the structure of the text and the extreme and erroneous interpretation of topics such as HIV and mortality rates, and conversely, the content of the article to be risky as it promotes fear rather than understanding of Christian sustainability.

Also, please understand that there is a lot of room for revision, given the fact that there is a lot of data and advanced research cases unique to Korea, and we are rejecting this article in a very positive light.

Q0. The question on arguing

One of the major challenges in this paper is that in discussing the significance of using the SIR model and the sustainability of Christianity, we believe that an in-depth discussion about why we should talk about it in the context of HIV and mortality is necessary.

We believe that they can create a false understanding, a simplistic interpretation. We also think that consideration should be given to the readers who search for and read this paper.

What is important in continuing to be a believer is the rhythm of one's life, external factors, social and economic factors, etc., and even when discussing the transmission model, it is not possible to speak only of opportunities for contact with people.

In addition, we cannot discuss the propagation model only from the perspective of opportunities for contact with people, because it depends on a variety of factors such as beliefs and communication with people we come into contact with on a daily basis, as well as external social factors (this paper uses newspapers as a case study, but I believe that personal media such as SNS may also have an influence on the model.)

In particular, the Republic of Korea should be an information-oriented country, since education and development in web media have advanced at a rapid pace since around 2000. For this reason, the influence of information via web media cannot be ignored.

Overall comment from section

Also, in discussing the sustainability of communication and religious communication in this paper, I think it is better to discuss the model case of COVID-19, where many risks in the church occurred. Various case studies and datasets have already been collected, and I suggest that the argument in this study may be more plausible in discussing the calculations of the SIR model used in this study and sustainability in religious beliefs if we use this logic again.

[1]Kim, B. N., Kim, E., Lee, S., & Oh, C. (2020). Mathematical Model of COVID-19 Transmission Dynamics in South Korea: The Impacts of Travel Restrictions, Social Distancing, and Early Detection. Processes, 8(10), 1304.

[2]Kim, Sungchan, et al. "Evaluation of COVID-19 epidemic outbreak caused by temporal contact-increase in South Korea. Infectious Diseases (2020).

[3]Feng, X., Chen, J., Wang, K., Wang, L., Zhang, F., Jin, Z., ... & Wang, X. (2020). Phase-adjusted estimation of the COVID-19 outbreak in South Korea under multi-source data and adjustment measures: a modelling study. Mathematical Biosciences and Engineering, 17(4), 3637.

In the above, a case study in a church is mentioned.

Other cases, such as a case study in COVID-19 with machine learning are discussed.

[4]Hong, H. G., & Li, Y. (2020). Estimation of time-varying reproduction numbers underlying epidemiological processes: A new statistical tool for the COVID-19 pandemic. PloS one, 15 (7), e0236464.

[5]Suzuki, Y., & Suzuki, A. (2020). Machine learning model estimating the number of COVID-19 infection cases over coming 24 days in every province of South Korea (XGBoost and MultiOutputRegressor). medRxiv. medRxiv.

[6]Althouse, B. M., Wallace, B., Case, B., Scarpino, S. V., Berdahl, A. M., White, E. R., & Hebert-Dufresne, L. (2020). The unintended consequences of inconsistent pandemic control policies. arXiv preprint arXiv:2008.09629.

In the above, the case of COVID-19 is used as a case study regarding the risk of infection in a densely populated zone, such as inside a church, where it was not expected.

Also, in arguing for the SIR model with COVID-19 as its axis, we think it is necessary to discuss the security of spatial distance in the current environment, such as Social Distanse, in order to make a final argument for the SIR model.

[7]Ho, Y. C., Chen, Y. H., Hung, S. H., Huang, C. H., Po, P., Chan, C. H., ... & Fang, C. T. (2020). Social Distancing 2.0 with Privacy-Preserving Contact Tracing to Avoid a Second Wave of COVID-19. arXiv preprint arXiv:2006.16611.

The above also raises issues regarding zoning in churches and other faiths.

[8]Bae, T. W., Kwon, K. K., & Kim, K. H. (2020). Mass Infection Analysis of COVID-19 Using the SEIRD Model in Daegu-Gyeongbuk of Korea from April to May, 2020. Journal of Korean medical science, 35(34).

In the above, a case study is described in which a model calculation was performed on practical responses to infection cases in hospitals and via churches.

In addition, previous studies on the risk of infection in Christian churches in SARS and MARS, etc., provide insufficient explanation to arrive at a logic from HIV and mortality, or a logic from evidence to sustainability in the faith, and yet, the explanatory variables used in the model calculations are Insufficient disclosure of the original data of text mining results.

The current research paper discusses the results of model calculations related to the risk of infection and mortality related to HIV infection and mortality as a case study using the SIR model with respect to sustainability in long-term Christian beliefs, but in discussing medium- and long-term sustainability, it is of course important to note that there is a lack of disclosure of the original data related to the risk of infection to wounds and diseases. Although there is a great need to be aware of the issues, there is no concrete explanation of the reasons for focusing only on HIV and mortality. In addition, it would be possible to discuss the issue from the perspectives of medical statistics and actuarial science and welfare.

In terms of consideration for the reader, I believe that the readers are pastors and others who work in the field of sustainability in the Christian faith.

We believe that considerations related to HIV management and pathology are more practical and easier to understand if you raise issues related to the risk of viral infection, such as the response to COVID-19, or if you take a case study approach to fitting a model.

However, although the above is understandable when considered in the context of Christian hospital management, it is still a limited story, and as a case study in discussing the sustainability of faith in model calculations using the SIR model, it is a bit extreme and should be considered from the reader's perspective. We think.

In particular, we assume that the reader is a practicing pastor, nun, or religious person.

With that in mind, we think it would be a negative campaign to discuss the sustainability of faith in the context of HIV, mortality, etc. in this paper. If we are going to discuss sustainability, we need a logic about comprehensive care in the context of the current crisis situation, such as COVID-19.

Q1.FIG: Regarding the diagram

Table 2 and figures should be converted to graphs and pivot tables.

Table 2, in particular, is very difficult to read with only numbers. We would like to see the contents of the data set graphed with consideration for the reader.

Q2. Data sets

There is no way to validate the model calculations as there is no description of how the dataset was obtained, the duration of the data, or what percentage of newspaper articles in Korea are influence factors by coverage.

When incorporating text mining results as an external factor into the model of the analysis, an exact description of the dataset is required since the results are estimates, taking into account only the influence on limited textual information. In addition, explicitly stated text mining rules, such as morphological analysis of parameters in model calculations, methods in text mining, program languages (or toolkits), and criteria for decision making when segmentation is used, will further enhance understanding of the explanatory factors and the SIR model , the significance of using the SD method, and understanding of the results will be promoted.

Q3. Regarding the text

Line; 110-112

Lines; 360-366

The decline of faith in scandals has occurred in many Christian denominations, especially in cases where the separation of church and state is not possible and the issue is obscured and not faced up to.

In recent years, the denunciation of Christianity has been questioned by social networking sites, dramas, music, movies, and other media that contain a vast amount of information about Christianity. In recent years, external accusations have been made on public message boards, which sometimes undermine the credibility of Christianity.

Especially in Korea, an information-oriented country, there are already many strategies in place.

There are risks lurking there as well, and I think there are cases where they may be deliberately destroyed over the medium to long term by the medium of content and other large scale information resources.

Although people have freedom of religion, it is possible that such large-scale information dissemination could lead to historical and cultural destruction (such as stigma and other destructive factors) in areas where the separation of church and state is difficult to achieve due to regional attributes and class balance.

We would have liked to see some reference in this paper to positive and negative effects on different axes than population trends. As a factor, depending on the directionality of the information and contents of the mass and personal media that we see in our daily lives, they may function to maintain the cultural image and image recognition of the region over the medium to long term.

We propose that the impact of external factors on large scale digital data, such as social networking sites and other uniquely advanced Korean digital data as external factors: the text of elements such as WOM (word-of-mouth information) and their analysis and scoring using topical models such as LDA can be further interpreted by utilizing the results of the analysis and scoring of these elements.

We believe that a guided and correct contextual discourse that leads to a faith that is understandable to the reader will build a sustainable society.

Reviewer #2: This paper has a lot of potential in contributing to the burgeoning literature on church growth/decline using SD and ABM approaches. However, there are a lot of issues that need to be sorted out first. A major problem is that the manuscript is difficult to follow, and this is primarily because the framing is not focused on the actual topic at hand. The authors tend to conflate the broad concept of sustainability, “environment”, and the topic of church membership growth, decline, and equilibrium. It took me about five pages to confirm that the paper would primarily focus on church membership, and not be tackling the many ways that religion is associated with social issues like environmental sustainability, or business concerns like financial sustainability. While there is an argument to be made that these are all linked to church membership, to an extent, it is distracting to the purpose of the paper.

Therefore, I strongly recommend re-framing the article to focus on church membership growth. Sentences such as “the extent to which churches contribute to sustainability in a society may drive the growth of the church population” Pg 27, lines 569-570, should instead read something closer to: Church contributions and involvement in bettering society help improve the church’s image, making people more open to receiving their evangelism efforts. “Environmental issues” also don’t rank highly in the reasons for declining church membership (claimed by the authors on lines 112/113). Reasons for decline are more varied than just ethical leadership and concern with social/environment issues. Much of it is demographic (births), which Hayward (2005, 2018) notes. The authors could spend more time discussing how births are handled in the model. It appears that new births are considered unbelievers, although it seems more likely that children will be similar to their parents in terms of religious belief; it would be nice if the authors could confirm they are following previous models’ standards in regards to births.

It is also unclear whether the main contribution is methodological, substantive, or comparative. Is it using an agent-based framework instead of just the system dynamics? The substantive results? Applying previously-designed models of church growth to the South Korean context? From my reading, the unique contribution seems to be using a time-dependent sustainable potential, which is informed by an analysis of newspapers. If this is true, then attention should be shifted to that in particular. Much more detail needs to be provided to defend the selection of sustainability indicators on page 24, which cites 7 articles, but does not elaborate on how they informed the decision-making: “the sustainability indicators of a church were selected from scholarly works related to practices for 508 sustainability [5,9,18,25,62-64].” For reasons discussed earlier, it isn’t clear to me why “environmental” would be its own category. A Church’s engagement in environmental issues seems similar to their engagement in other issues that are important to the broader public.

More detailed suggestions are as follows:

1. The paper would benefit from more background information specific to church growth/decline and about the Presbyterian Church in Korea. How relevant is discussion of revival, like the Great Revival in South Korea, to modeling decisions with SD and ABM approaches? How does it inform the selection of indicators in the sustainability function?

2. The line charts (Figure 4) are difficult to read

3. Be careful with generalizations to “religion” when aspects of Christianity or Presbyterianism are the focus. The term “biblical worldview” for instance (line 33). Some of the statements about religion are also unnecessarily broad (e.g. sentence starting at line 44); keep it focused on church membership changes.

4. “Pippa” should be “Norris” (line 51). Unclear that this citation supports the argument anyways, since they assert that religiosity tends to decrease with increased socioeconomic well-being.

5. “In the discipleship process, passive believers can mature into active believers, where the process usually lasts four years in church situations in South Korea” pg 12 lines 252-253 – are there any sources to support this claim?

6. “Initial parameters of flows and other auxiliary variables were optimized to reproduce the trend of the PCK population from 1995 to 2018” pg 20 lines 422-423 -- How is this optimization completed?

Overall, I view re-framing of the paper to focus on church growth/decline, instead of “sustainability” to be of the utmost importance. Besides that, however, there needs to be more work done to make the paper a clearer contribution, with more detailed explanation of the sustainability function in particular. This may be a challenge since religion does not appear to be the authors’ usual field of study. I would advise them to review the citations in Hayward’s 2018 paper to become more acquainted with the study of church growth/decline, which has primarily focused on the United States setting. If the authors are successful, I would certainly look forward to reading the paper again.

6. PLOS authors have the option to publish the peer review history of their article (what does this mean?). If published, this will include your full peer review and any attached files.

Reviewer #1: No

Reviewer #2: No

---

## [Author Response · Author response to Decision Letter 0]

15 Feb 2021

PONE-D-20-30060

Church population dynamics with sustainability

PLOS ONE

Dear Editor,

Thank you for inviting us to submit a revised draft of our manuscript entitled, "Church population dynamics with sustainability" to PLOS ONE. We also appreciate the time and effort you and each of the reviewers have dedicated to providing insightful feedback on ways to strengthen our paper. Thus, it is with great pleasure that we resubmit our article for further consideration. We have incorporated changes that reflect the detailed suggestions you have graciously provided. We also hope that our edits and the responses we provide below satisfactorily address all the issues and concerns you and the reviewers have noted.

To facilitate your review of our revisions, the following is a point-by-point response to the questions and comments.

Again, thank you for giving us the opportunity to strengthen our manuscript with your valuable comments and queries. We have worked hard to incorporate your feedback and hope that these revisions persuade you to accept our submission.

Sincerely,

Jae-Woo Lee

Editor suggestions:

In preparing your revision to address all comments by the two reviewers, I hope you would give special attention to: (1) your choice of title for the manuscript, since its scope is significantly narrower than the question on sustainability of churches; (2) the framing of your study, as both reviewers feel that the present perspective is rather negative, but at the same time, there is tremendous potential to use the mathematical modeling more positively. 

RESPONSE: The authors change the title to “Sustainability of Religious Population” and reframe overall contents in the responses to Reviewer #2.

Journal Requirements:

2. In your Data Availability statement, you have not specified where the minimal data set underlying the results described in your manuscript can be found. PLOS defines a study's minimal data set as the underlying data used to reach the conclusions drawn in the manuscript and any additional data required to replicate the reported study findings in their entirety. All PLOS journals require that the minimal data set be made fully available.

RESPONSE: The authors provide study’s minimal underlying data set as either Supporting Information files to be able to replicate the reported study.

Reviewers' comments:

Reviewer #1: Overall comment

We are very sorry to have to reject it this time, but we think it would be a tremendous improvement if you could make better use of the Korean original data set and present a data set that can be used for empirical experiments.

RESPONSE: The authors use original data provided by the National Statistical Information Service on the birth rate and mortality rate of the Korean population [56]. The results of the simulated church population using this data are compared to the original data provided by the Korean Presbyterian Church denomination [33]. We provide the Korean original data set as Supporting Information files and discuss in depth.

56. Korean Statistical Information Service [Internet]. 2020 National demographics [cited 2019 Dec 10]. Available from: https://www.kosis.kr

33. The Presbyterian Church of Korea [Internet]. The annual reports of the presbyterian church of Korea general assembly [cited 2020 Mar 11]. Available from: http://www.pck.or.kr/

We also agree with the theme of the paper and the ideology of the abstract.

RESPONSE: We appreciate for your agreement to the theme and the ideology. 

However, in agreeing with you, we find the structure of the text and the extreme and erroneous interpretation of topics such as HIV and mortality rates, and conversely, the content of the article to be risky as it promotes fear rather than understanding of Christian sustainability.

Response: We adopt the HIV model as an SIR model to interpret the dynamics of religious people. Here the SIR corresponds to UAB (U=unbelievers, A=active believers, B=inactive believers). It is not out aim to make fears for Christian sustainability. Conversely, out aim is to guide to promote the sustainability of the Christian. If someone understand the underlying dynamics of changing situation, they should prepare to adapt to the new environments.

Also, please understand that there is a lot of room for revision, given the fact that there is a lot of data and advanced research cases unique to Korea, and we are rejecting this article in a very positive light.

Response: We modified the article following your helpful comments.

Q0. The question on arguing

One of the major challenges in this paper is that in discussing the significance of using the SIR model and the sustainability of Christianity, we believe that an in-depth discussion about why we should talk about it in the context of HIV and mortality is necessary.

RESPONSE: For mortality, the simple church population SD model assumes that birth rate and mortality rate can be ignored when considering short-term church growth of about 15 years. However, the extended models that include long-term effects over a generation should take into account birth rates and mortality rates [46].

46. Hayward J. A general model of church growth and decline. Journal of Mathematical Sociology. 2005 Jul 1;29(3):177-207.

For context of HIV, the transmission from the susceptible A state to the infected B state by contacts with an infected agent can be well described by the SIR approach [Bicher, 2015]. The S-shaped population growth appears throughout multiple fields of science [Zelinka, 2019] and is used to model epidemics by the SIR approach, where the fundamental assumption of population growth is that transmission results from contact between susceptible individual and infected individual. The SIR approach well explains the transmission of HIV infection [Zakary, 2016]. The authors use the HIV ABM model to describe the church population growth with S-shaped growth archetype upon the SIR approach. 

Bicher, M., Glock, B., Miksch, F., Popper, N., & Schneckenreither, G. (2015). Definition, validation and comparison of two population models for Austria. International Journal of Business and Technology, 4(1), 7.

Zelinka, D., & Amadei, B. (2019). A systems approach for modeling interactions among the Sustainable Development Goals Part 2: System dynamics. International Journal of System Dynamics Applications (IJSDA), 8(1), 41-59.

Zakary, O., Rachik, M., & Elmouki, I. (2016). On the impact of awareness programs in HIV/AIDS prevention: an SIR model with optimal control. Int. J. Comput. Appl, 133(9), 1-6.

We believe that they can create a false understanding, a simplistic interpretation. We also think that consideration should be given to the readers who search for and read this paper.

What is important in continuing to be a believer is the rhythm of one's life, external factors, social and economic factors, etc., and even when discussing the transmission model, it is not possible to speak only of opportunities for contact with people.

In addition, we cannot discuss the propagation model only from the perspective of opportunities for contact with people, because it depends on a variety of factors such as beliefs and communication with people we come into contact with on a daily basis, as well as external social factors (this paper uses newspapers as a case study, but I believe that personal media such as SNS may also have an influence on the model.)

In particular, the Republic of Korea should be an information-oriented country, since education and development in web media have advanced at a rapid pace since around 2000. For this reason, the influence of information via web media cannot be ignored.

RESPONSE: The underlying assumption in the traditional SIR model is that the population is homogenous; that is, everybody makes contact with each other randomly. There is no latency, i.e., there is no time delay between exposure to the infection and actually getting infected. Essentially, susceptible individuals can acquire the infection through contacts with infectious individuals, and as soon as they are infected, they proceed to the infectious stage [27].

For church growth conversion is the equivalent of acquiring the infection. The fundamental assumption of conversion growth is that converts result from a typical contact between active believers and unbelievers on the SIR framework [26], although there are distinguishable contacts in various social networks. The authors assume that the church grows by contacts between active members and non-believers on a social network. The church population growth occurs through the conversion process in which active believers of the church members bring unbelievers into the church. 

Additionally, the authors used HIV ABM framework in which HIV caused by contacts with the infected agent is transmitted while applying the church population to the ABM method. Much of the work in ABM has focused on the simulation of contact-based infection spread associated with influenza-like illnesses, other respiratory infections and sexually transmitted infection. The intentional transmission mode of HIV is sexual contact, direct contact with HIV-infected blood or fluids. The simple HIV ABM framework limits the consideration of only typical agent contact without contact patterns in the real world, such as contact between family members, classmates, and co-workers [Liu, 2018]. 

The simple HIV ABM framework does not include ‘data culture’ to generate agent contact and movement patterns like intelligent transportation system, cellular service provider data, personal information on social media and technologies that may leverage smartphone and another mobile device [Rutherford, 2012]. 

The proposed ABM is a minimal model to understand the dynamics of changing Christians.

Therefore, the authors exclude social media contacts in this paper.

27. Hayward J. Mathematical modeling of church growth. The Journal of mathematical sociology. 1999 Feb 1;23(4):255-92.

26. Hayward J. Growth and decline of religious and subcultural groups. In Proceedings of the 18th International Conference of the System Dynamics Society 2000 Aug 6.

Liu, Q. H., Ajelli, M., Aleta, A., Merler, S., Moreno, Y., & Vespignani, A. (2018). Measurability of the epidemic reproduction number in data-driven contact networks. Proceedings of the National Academy of Sciences, 115(50), 12680-12685.

Rutherford, G., Friesen, M. R., & McLeod, R. D. (2012). An agent based model for simulating the spread of sexually transmitted infections. Online Journal of Public Health Informatics, 4(3).

Overall comment from section

Also, in discussing the sustainability of communication and religious communication in this paper, I think it is better to discuss the model case of COVID-19, where many risks in the church occurred. Various case studies and datasets have already been collected, and I suggest that the argument in this study may be more plausible in discussing the calculations of the SIR model used in this study and sustainability in religious beliefs if we use this logic again.

[1]Kim, B. N., Kim, E., Lee, S., & Oh, C. (2020). Mathematical Model of COVID-19 Transmission Dynamics in South Korea: The Impacts of Travel Restrictions, Social Distancing, and Early Detection. Processes, 8(10), 1304.

[2]Kim, Sungchan, et al. "Evaluation of COVID-19 epidemic outbreak caused by temporal contact-increase in South Korea. Infectious Diseases (2020).

[3]Feng, X., Chen, J., Wang, K., Wang, L., Zhang, F., Jin, Z., ... & Wang, X. (2020). Phase-adjusted estimation of the COVID-19 outbreak in South Korea under multi-source data and adjustment measures: a modelling study. Mathematical Biosciences and Engineering, 17(4), 3637.

In the above, a case study in a church is mentioned.

Other cases, such as a case study in COVID-19 with machine learning are discussed.

[4]Hong, H. G., & Li, Y. (2020). Estimation of time-varying reproduction numbers underlying epidemiological processes: A new statistical tool for the COVID-19 pandemic. PloS one, 15 (7), e0236464.

[5]Suzuki, Y., & Suzuki, A. (2020). Machine learning model estimating the number of COVID-19 infection cases over coming 24 days in every province of South Korea (XGBoost and MultiOutputRegressor). medRxiv. medRxiv.

[6]Althouse, B. M., Wallace, B., Case, B., Scarpino, S. V., Berdahl, A. M., White, E. R., & Hebert-Dufresne, L. (2020). The unintended consequences of inconsistent pandemic control policies. arXiv preprint arXiv:2008.09629.

In the above, the case of COVID-19 is used as a case study regarding the risk of infection in a densely populated zone, such as inside a church, where it was not expected.

Also, in arguing for the SIR model with COVID-19 as its axis, we think it is necessary to discuss the security of spatial distance in the current environment, such as Social Distanse, in order to make a final argument for the SIR model.

[7]Ho, Y. C., Chen, Y. H., Hung, S. H., Huang, C. H., Po, P., Chan, C. H., ... & Fang, C. T. (2020). Social Distancing 2.0 with Privacy-Preserving Contact Tracing to Avoid a Second Wave of COVID-19. arXiv preprint arXiv:2006.16611.

The above also raises issues regarding zoning in churches and other faiths.

[8]Bae, T. W., Kwon, K. K., & Kim, K. H. (2020). Mass Infection Analysis of COVID-19 Using the SEIRD Model in Daegu-Gyeongbuk of Korea from April to May, 2020. Journal of Korean medical science, 35(34).

In the above, a case study is described in which a model calculation was performed on practical responses to infection cases in hospitals and via churches.

In addition, previous studies on the risk of infection in Christian churches in SARS and MARS, etc., provide insufficient explanation to arrive at a logic from HIV and mortality, or a logic from evidence to sustainability in the faith, and yet, the explanatory variables used in the model calculations are Insufficient disclosure of the original data of text mining results.

The current research paper discusses the results of model calculations related to the risk of infection and mortality related to HIV infection and mortality as a case study using the SIR model with respect to sustainability in long-term Christian beliefs, but in discussing medium- and long-term sustainability, it is of course important to note that there is a lack of disclosure of the original data related to the risk of infection to wounds and diseases. Although there is a great need to be aware of the issues, there is no concrete explanation of the reasons for focusing only on HIV and mortality. In addition, it would be possible to discuss the issue from the perspectives of medical statistics and actuarial science and welfare.

In terms of consideration for the reader, I believe that the readers are pastors and others who work in the field of sustainability in the Christian faith.

We believe that considerations related to HIV management and pathology are more practical and easier to understand if you raise issues related to the risk of viral infection, such as the response to COVID-19, or if you take a case study approach to fitting a model.

However, although the above is understandable when considered in the context of Christian hospital management, it is still a limited story, and as a case study in discussing the sustainability of faith in model calculations using the SIR model, it is a bit extreme and should be considered from the reader's perspective. We think.

In particular, we assume that the reader is a practicing pastor, nun, or religious person.

With that in mind, we think it would be a negative campaign to discuss the sustainability of faith in the context of HIV, mortality, etc. in this paper. If we are going to discuss sustainability, we need a logic about comprehensive care in the context of the current crisis situation, such as COVID-19.

RESPONSE: The authors appreciate that the reviewer is providing valuable insight into the relevance between the model case of COVID-19 and the sustainability of church population. A valid contact between active believers and non-believers is an encounter that convinces non-believers to repent. The positive atmosphere that non-believers have toward the church makes contact effective.

In general, self-quarantine, social distancing and shelter in place are conducted in order to prevent the transmission of COVID-19. Problems arise when an infected believer participates in a church meeting. Korean churches have been criticized for the spread of the COVID-19 virus caused by the infected believers in contact with non-believers. This will negatively affect the long-term sustainability of the church by giving it a negative image even after the COVID-19 pandemic. Also, the situation in which the church is in conflict with the quarantine authorities has a negative impact on the growth of the church.

In the SIR or modified SIR model for COVID-19, this phenomenon is described by time-varying reproduction number [Hong, 2020]. In our present study, the time-dependent sustainable potential represents the time-varying reproduction number. We produce the time-dependent sustainable potential using text mining techniques for newspaper articles over years. However, the value is inappropriate to monitor changes over a short period of time. As the reviewer suggested, we agree that prompt information extracted from SNS communication is suitable for predicting the time-dependent sustainable potential influencing the sustainability of church population in the COVID-19 pandemic. In future studies, we can discuss the sustainability of church population on the COVID-19 SIR SD framework using the time-dependent sustainable potential obtained from SNS communication.

Hong, H. G., & Li, Y. (2020). Estimation of time-varying reproduction numbers underlying epidemiological processes: A new statistical tool for the COVID-19 pandemic. PloS one, 15(7), e0236464.

Q1.FIG: Regarding the diagram

Table 2 and figures should be converted to graphs and pivot tables. Table 2, in particular, is very difficult to read with only numbers. We would like to see the contents of the data set graphed with consideration for the reader.

RESPONSE: We changed Table 2 to a graph as shown Fig 7 and explained the contents of data set. Fig 7 shows the total number of newspaper articles searched from 1995 to 2018 using 22 sustainability indicators as keywords in BIGKinds system. The church-related sustainability indicators vary from a minimum of 596 to a maximum of 5591, but they are not concentrated in a specific indicator and exhibit an even distribution. The vertical width of each year segment shows when the corresponding sustainability indicator was of great social interest. For example, at the financial dimension of the church, the (1) donation indicator was 499 in 2015 year, showing a higher interest compared to other years. At the political dimension, (18) social justice was 337 in 2014 year, indicating that the interest of social justice was higher than that of other years.

Q2. Data sets

There is no way to validate the model calculations as there is no description of how the dataset was obtained, the duration of the data, or what percentage of newspaper articles in Korea are influence factors by coverage.

RESPONSE: The model validation was performed comparing the model’s results with the historical data. A reference for the validation-process of the church population growth SD model is obtained by the total population data of the Presbyterian Church of Kore (PCK) for 1995 - 2018. We compared the PCK data with the simulation results of the SD model for validation. The Mean Absolute Percentage Error (MAPE) metric is used to validate the SD model and the threshold for acceptable MAPE values is 10%. The MAPE is represented by following equation;

MAPE=1/N ∑_(t=1)^N▒〖(|A_t-F_t |)/A_t ×100〗

where A_t is the actual value and F_t is the forecasted value at time t, and N is number of periods. For the baseline scenario for s = 1 (and the referenced data set) MAPE=4.4 % and an R^2 value of 0.85 are observed during validation, which implies reasonable measure for the SD model [Lim, 2005].

[52] Lim, C. W., & Kirikoshi, T. (2005). Predicting the effects of physician-directed promotion on prescription yield and sales uptake using neural networks. Journal of Targeting, Measurement and Analysis for Marketing, 13(2), 156-167.

When incorporating text mining results as an external factor into the model of the analysis, an exact description of the dataset is required since the results are estimates, taking into account only the influence on limited textual information. In addition, explicitly stated text mining rules, such as morphological analysis of parameters in model calculations, methods in text mining, program languages (or toolkits), and criteria for decision making when segmentation is used, will further enhance understanding of the explanatory factors and the SIR model, the significance of using the SD method, and understanding of the results will be promoted.

RESPONSE: In order to determine the sustainable potential parameter in the SD model, the authors applied the external factor value measured by text mining technique for the sustainability of church in South Korean society. 22 sustainability indicators related to the sustainability of church in five areas: economy, education, society, politics, and environment were determined through literature review.

Figure 7 shows the total number of newspaper articles searched from 1995 to 2018 using the 22 sustainability indicators as keywords in BIGKinds system [64]. In order to apply news big data, which is an unstructured text, as analysis data, the data must be processed so that Natural Language Processing (NLP) can be performed. For data cleaning, the authors perform morphological analysis using machine learning and morpheme analysis dictionaries, especially using noun dictionaries built inside the BIGKinds system. The BIGKinds system uses a structured Support Vector Machine (SVM) algorithm for morphological analysis and data preprocessing. Automatically extract all noun keywords from the article text and remove stopwords. The structured SVM algorithm is a machine learning algorithm for NLP of text and shows 97.13% performance in Korean predicate recognition and classification.

[64] BIGKinds Korea Press Foundation [Internet]. News bigdata and analysis. Available from: http://www.kinds.or.kr. Accessed Feb 4, 2020.

Q3. Regarding the text

Line; 110-112

Lines; 360-366

The decline of faith in scandals has occurred in many Christian denominations, especially in cases where the separation of church and state is not possible and the issue is obscured and not faced up to.

In recent years, the denunciation of Christianity has been questioned by social networking sites, dramas, music, movies, and other media that contain a vast amount of information about Christianity. In recent years, external accusations have been made on public message boards, which sometimes undermine the credibility of Christianity.

Especially in Korea, an information-oriented country, there are already many strategies in place.

There are risks lurking there as well, and I think there are cases where they may be deliberately destroyed over the medium to long term by the medium of content and other large scale information resources.

Although people have freedom of religion, it is possible that such large-scale information dissemination could lead to historical and cultural destruction (such as stigma and other destructive factors) in areas where the separation of church and state is difficult to achieve due to regional attributes and class balance.

We would have liked to see some reference in this paper to positive and negative effects on different axes than population trends. As a factor, depending on the directionality of the information and contents of the mass and personal media that we see in our daily lives, they may function to maintain the cultural image and image recognition of the region over the medium to long term.

We propose that the impact of external factors on large scale digital data, such as social networking sites and other uniquely advanced Korean digital data as external factors: the text of elements such as WOM (word-of-mouth information) and their analysis and scoring using topical models such as LDA can be further interpreted by utilizing the results of the analysis and scoring of these elements.

We believe that a guided and correct contextual discourse that leads to a faith that is understandable to the reader will build a sustainable society.

RESPONSE: Social media-based relationships can lead to positive or negative word of mouth (WOM) recommendations. Social media interactions between believers and non-believers affect how non-believers think and feel about the church, and consequently, these interactions can lead to the positive WOM, affecting the conversion of non-believers.

Believers who access faith-based content on Facebook respond to liking, commenting, and sharing when faithful resources such as articles, sermons, and praises are posted. They can deliver spiritual enlightenment and entertainment to believers or non-believers connected on the Facebook network, and provide relaxation and enjoyment, creating a positive image of Christianity [Brubaker, 2017]. 

Believers can expect the growth of church through the advantages of Facebook, but they should discern and use posts considering the disadvantages of Facebook such as alias users, multi-account users, falsified information, the duplicity of information and the negative publicity [Kgatle, 2018]. Therefore, the social media usage gap between congregations rich in information resources and restricted in information resources should be narrowed [Lee, 2018].

Christian congregations use their Facebook to engage in non-religious activities, both in social service and in political activities. The Christian congregation's adoption of social media platforms is closely related to community service and social marketing activities and resources [Lee, 2018]. This indicates that in secular society Christianity is not only an inherited and maintained canopy, but also a cultural framework that generates public discourse online [Coman, 2017].

In addition, the image of a religious leader can appear positively or negatively depending on the method of communication in the relationship with the mass media [Baffelli, 2007]. Some people on social media are struggling with religious beliefs and convictions due to the pastor's non-Christian behavior. The Christianity such as sacrifice for others, humility, kindness, and altruism represents a major cultural frame for evaluating certain situations, for attributing the blame and imposing behavior norms [Coman, 2017]. Church leaders should understand this phenomenon and use social media as another social group in which believers should reflect on their daily actions and thoughts, and build a new public image.

The authors appreciate providing the insight for the religious role of social media. We can find and measure external factors that affect the sustainability of the church population appearing on social media, and reflect them in the next study.

Baffelli, E. (2007). Mass Media and Religion in Japan: Mediating the Leader's Image. Westminster Papers in Communication & Culture, 4(1).

[66] Brubaker, P. J., & Haigh, M. M. (2017). The religious Facebook experience: Uses and gratifications of faith-based content. Social Media+ Society, 3(2), 2056305117703723.

[67] Kgatle, M. S. (2018). Social media and religion: Missiological perspective on the link between Facebook and the emergence of prophetic churches in southern Africa. Verbum et Ecclesia, 39(1), 1-6.

[68] Lee, Y. J. (2018). Is your church “liked” on Facebook? Social media use of Christian congregations in the United States. Nonprofit Management and Leadership, 28(3), 383-398.

[69] Coman, I. A., & Coman, M. (2017). Religion, popular culture and social media: the construction of a religious leader image on Facebook. ESSACHESS–Journal for Communication Studies, 10(2 (20)), 129-143.

Reviewer #2: This paper has a lot of potential in contributing to the burgeoning literature on church growth/decline using SD and ABM approaches. However, there are a lot of issues that need to be sorted out first. A major problem is that the manuscript is difficult to follow, and this is primarily because the framing is not focused on the actual topic at hand. The authors tend to conflate the broad concept of sustainability, “environment”, and the topic of church membership growth, decline, and equilibrium. It took me about five pages to confirm that the paper would primarily focus on church membership, and not be tackling the many ways that religion is associated with social issues like environmental sustainability, or business concerns like financial sustainability. While there is an argument to be made that these are all linked to church membership, to an extent, it is distracting to the purpose of the paper.

Therefore, I strongly recommend re-framing the article to focus on church membership growth. Sentences such as “the extent to which churches contribute to sustainability in a society may drive the growth of the church population” Pg 27, lines 569-570, should instead read something closer to: Church contributions and involvement in bettering society help improve the church’s image, making people more open to receiving their evangelism efforts. “Environmental issues” also don’t rank highly in the reasons for declining church membership (claimed by the authors on lines 112/113). Reasons for decline are more varied than just ethical leadership and concern with social/environment issues. Much of it is demographic (births), which Hayward (2005, 2018) notes. The authors could spend more time discussing how births are handled in the model. It appears that new births are considered unbelievers, although it seems more likely that children will be similar to their parents in terms of religious belief; it would be nice if the authors could confirm they are following previous models’ standards in regards to births.

RESPONSE: The authors reconstruct the content of the paper on how the growth of the church population and the sustainability of the church can be linked and integrated. 

We re-frame the introduction part in line 59-63 of page 4. As you suggested, we introduce the contribution of churches to sustainability in a society.

The theory of church growth is concerned with social factors that strengthen or inhibit growth, or with factors within the church itself. Another source of church growth is in retaining the children of believing parents. The generalized church growth model of Hayward assumes that such children are included in the number of churches from birth. So, the church population growth model considering more than one generation should take into account birth rates and mortality rates, along with predicting social factors [45].

Following paragraph appear in line 85-98 of page 5. 

The 2015 Population and Housing Census in South Korea showed an increase in the Protestant population, but this was not because the church was healthy or evangelized eagerly but due to the cumulative effect of the Protestant population. This is because parents bring their children to church more than other religious groups. In fact, if the cumulative effect is removed, the Protestant population declined by more than 10% [Newnjoy, 2017]. The reason for the decline in the Korean church is due to the political corruption of Christians and the negative social image of the church. Bad incidents against Christianity have made a negative impression on the public and Christianity is recognized as a religion of division among the public.

In a poll on 2018, the ‘ethics and practice movement’ (45.5%) was cited as the top priority for enhancing the credibility of the Korean church. Volunteer and relief activities recorded the second place with 36.4 % [Christian Newspaper, 2018]. This shows that church contributions and involvement in bettering society help improve the church’s image, making people more open to receiving their evangelism efforts.

45. Hayward J. Mathematical modeling of church growth: A system dynamics approach. arXiv preprint arXiv:1805.08482. 2018 May 22.

Newnjoy, (2017). http://www.newsnjoy.or.kr/news/articleView.html?idxno=208019. 

Christian Newspaper, (2018). http://www.gdknews.kr/news/view.php?no=2675.

It is also unclear whether the main contribution is methodological, substantive, or comparative. Is it using an agent-based framework instead of just the system dynamics? The substantive results? Applying previously-designed models of church growth to the South Korean context? 

RESPONSE: The authors showed that the sustainable potential parameter in the prototype SD model and the simple ABM model for church population growth plays an important role in achieving the stability of church population. 

The church population SD model on SIR approach is driven by contacts with believers and non-believers, and the number of contacts is adjusted to the conversion parameter. However, it is not easy to understand what the change of this parameter actually means over specific time changes. On the other hand, the ABM framework on the SIR approach, which is implemented by the contact of agents, provides a micro perspective to understand changes in the number of contacts over specific time changes. We investigated the role of the disciple rate for church members in the ABM framework, because the ABM approach is useful for understanding the effect of time changes. Using the ABM framework, we confirm that the period of discipleship training is shortened to four, three, and two years, which rather interfere with the growth of the church as the active believers increase.

From my reading, the unique contribution seems to be using a time-dependent sustainable potential, which is informed by an analysis of newspapers. If this is true, then attention should be shifted to that in particular. Much more detail needs to be provided to defend the selection of sustainability indicators on page 24, which cites 7 articles, but does not elaborate on how they informed the decision-making: “the sustainability indicators of a church were selected from scholarly works related to practices for 508 sustainability [5,9,18,25,62-64].” For reasons discussed earlier, it isn’t clear to me why “environmental” would be its own category. A Church’s engagement in environmental issues seems similar to their engagement in other issues that are important to the broader public.

RESPONSE: The authors decided on 22 representative indicators in the five sustainability dimensions consisting of economic, educational, social, political and environmental domains as defined in Fig 6. Sustainability indicators refer to social environmental factors and elements within the church that directly and indirectly affect the sustainability of church.

Among sustainability indicators, the church's financial income is collected by voluntary donation, bazaar fundraising, and personal fundraising or offering. Church education takes place individually through counseling and collectively through Sunday School education, and the expenditure of scholarship in church finance plays a role in promoting church education by supporting seminary students. The church's social activities are carried out by volunteers who deliver relief supplies to the elderly, the disabled, the poor, single parents, and the victims of the disaster. The political activities of church are conducted by participation in public hearings and elections to realize social justice. The church's interests in the field of environment come from the creation faith and appear in all environmental movements aimed at preserving the nature that God created.

More detailed suggestions are as follows:

1. The paper would benefit from more background information specific to church growth/decline and about the Presbyterian Church in Korea. How relevant is discussion of revival, like the Great Revival in South Korea, to modeling decisions with SD and ABM approaches? How does it inform the selection of indicators in the sustainability function?

RESPONSE: The ABM-approaching church population goes through an S-shaped curve with exponential growth and reaches stabilization in 25 years for s = 2 and 17 years for s = 3 in Fig 3. This is in good agreement with the results of the SD approach in Fig 2. The exponential growth of the S-shaped curve means the great revival of church. This result indicates that the sustainable potential s must be above the threshold in order to achieve continuous stabilization after the great revival inside one generation.

2. The line charts (Figure 4) are difficult to read

RESPONSE: The authors revise Fig 4 of the extended SD model to make it easier to understand.

3. Be careful with generalizations to “religion” when aspects of Christianity or Presbyterianism are the focus. The term “biblical worldview” for instance (line 33). Some of the statements about religion are also unnecessarily broad (e.g. sentence starting at line 44); keep it focused on church membership changes.

RESPONSE: The authors expelled the term of “biblical” and evicted a broad discussion of religion and again described the content with a focus on Christianity. 

4. “Pippa” should be “Norris” (line 51). Unclear that this citation supports the argument anyways, since they assert that religiosity tends to decrease with increased socioeconomic well-being.

RESPONSE: The authors rechecked the references and described them accurately.

5. “In the discipleship process, passive believers can mature into active believers, where the process usually lasts four years in church situations in South Korea” pg 12 lines 252-253 – are there any sources to support this claim?

RESPONSE: The authors presented accurate reference to the above passage: “In the discipleship process, passive believers can mature into active believers, where the process usually lasts four years in church situations in South Korea [50]”

50 Chai YG, Chai D. A New Testament church in the 21st century: The house church. GLPI; 2010.

6. “Initial parameters of flows and other auxiliary variables were optimized to reproduce the trend of the PCK population from 1995 to 2018” pg 20 lines 422-423 -- How is this optimization completed?

RESPONSE: The authors conducted the optimization test called as the one-at-a-time method to fix the values of other variables when optimizing the value of one parameter [Kuai, 2015]. The optimization test is basically allowing to adjust the parameters involved in the model so it produces a minimal error according to some error metric on a specific test dataset [Khurram, 2015]. We determine the set of parameters by comparing simulation results with reference data, so that the difference between simulation output and reference data is as small as possible [Bicher, 2017].

The optimization process is to verify the validity of the model by comparing the simulated result and the real values. The model validation was performed comparing the model’s results with the historical data. A reference for the validation-process of the church population growth SD model is obtained by the total population data of the Presbyterian Church of Kore (PCK) for 1995 - 2018. We compared the PCK data with the simulation results of the SD model for validation. The Mean Absolute Percentage Error (MAPE) metric is used to validate the SD model and the threshold for acceptable MAPE values is 10%. The MAPE is represented by following equation;

MAPE=1/N ∑_(t=1)^N▒〖(|A_t-F_t |)/A_t ×100〗

where A_t is the actual value and F_t is the forecasted value at time t, and N is number of periods. For the baseline scenario for s = 1 (and the referenced data set) MAPE=4.4 % and an R^2 value of 0.85 are observed during validation, which implies reasonable measure for the SD model [Lim, 2005].

Kuai, P., Li, W., & Liu, N. (2015). Evaluating the effects of land use planning for non-point source pollution based on a system dynamics approach in China. PloS one, 10(8), e0135572.

Khurram Jassal, M. (2011). The Effect of Optimization of Error Metrics.

Lim, C. W., & Kirikoshi, T. (2005). Predicting the effects of physician-directed promotion on prescription yield and sales uptake using neural networks. Journal of Targeting, Measurement and Analysis for Marketing, 13(2), 156-167.

Bicher, M., Urach, C., Zauner, G., Rippinger, C., & Popper, N. (2017, December). Calibration of a stochastic agent-based model for re-hospitalization numbers of psychiatric patients. In 2017 Winter Simulation Conference (WSC) (pp. 2940-2951). IEEE..

Overall, I view re-framing of the paper to focus on church growth/decline, instead of “sustainability” to be of the utmost importance. Besides that, however, there needs to be more work done to make the paper a clearer contribution, with more detailed explanation of the sustainability function in particular. This may be a challenge since religion does not appear to be the authors’ usual field of study. I would advise them to review the citations in Hayward’s 2018 paper to become more acquainted with the study of church growth/decline, which has primarily focused on the United States setting. If the authors are successful, I would certainly look forward to reading the paper again.

RESPONSE: The authors change the title to “Sustainability of Religious Population” and reframe overall contents focusing on sustainability function and the Korean church population growth/decline based on above responses.

In this paper, the function of sustainability is represented by the sustainable parameter. Raising the sustainable parameter is a way to achieve the sustainability of the church population because the church population stabilizes when the sustainable parameter exceeds the threshold. However, the discussion on how to increase the sustainable parameter by adjusting 22 sustainability indicators goes beyond the scope of this paper, as it is expected to undergo complex processes requiring big data analysis. 

On the other hand, we propose that increasing the conversion fraction can be an alternative to achieving sustainability when the sustainable parameter fails to exceed the threshold. This is achieved by increasing contacts between believers and non-believers.

---

## [Decision Letter · Decision Letter 1]

16 Mar 2021

PONE-D-20-30060R1

Sustainability of Riligious Population

PLOS ONE

Dear Dr. Lee,

Thank you for submitting your manuscript to PLOS ONE. After careful consideration, we feel that it has merit but does not fully meet PLOS ONE’s publication criteria as it currently stands. Therefore, we invite you to submit a revised version of the manuscript that addresses the points raised during the review process.

Please address the remaining concerns of Reviewer 2. Also, 'Riligious Population' sounds awkward in the title, and 'Religious' is misspelled. Will the authors consider 'Religious Communities' instead?

We look forward to receiving your revised manuscript.

Kind regards,

Siew Ann Cheong, Ph.D.

Academic Editor

PLOS ONE

Journal Requirements:

Reviewers' comments:

Reviewer's Responses to Questions

**Comments to the Author**

1. If the authors have adequately addressed your comments raised in a previous round of review and you feel that this manuscript is now acceptable for publication, you may indicate that here to bypass the “Comments to the Author” section, enter your conflict of interest statement in the “Confidential to Editor” section, and submit your "Accept" recommendation.

Reviewer #1: All comments have been addressed

Reviewer #2: (No Response)

2. Is the manuscript technically sound, and do the data support the conclusions?

Reviewer #1: Partly

Reviewer #2: Yes

3. Has the statistical analysis been performed appropriately and rigorously? 

Reviewer #1: Yes

Reviewer #2: I Don't Know

4. Have the authors made all data underlying the findings in their manuscript fully available?

Reviewer #1: Yes

Reviewer #2: Yes

5. Is the manuscript presented in an intelligible fashion and written in standard English?

Reviewer #1: Yes

Reviewer #2: No

6. Review Comments to the Author

Reviewer #1: Most of the points I commented on before have been improved and the paper is now very easy to read and understand.

In particular, it is easier to understand the allocation of topics, the position of the research in the timeline, and the correspondence and relationship with previous research in Korea.

Reviewer #2: I appreciate the efforts the authors put into responding to my earlier comments. The removal of text in the first few pages and addition of other text throughout does improve the framing. The focus now seems more about the public image and functional purpose of the church in Korean society, and how improving these assists with membership sustainability by making the un-churched more open to joining the church. A key phrase in helping me understand this was on page 12: “Sustainable potential s reflects how amicable unbelievers are to the Christian faith, and integrates social factors in a society.” This contrasts to previous approaches, which were focused on conversion potential via the level of enthusiasm for evangelism among current members.

Unfortunately, the contribution of the sustainability potential still doesn’t become clear until nearly half-way through the paper. There is still a problem that sustainability is introduced in a way that many readers may find confusing. The introduction should be about the sustainability of church membership through many avenues including conversion efforts, not how religious values or beliefs are related to the field of sustainability. The first two paragraphs should be written to highlight the way that past literature tends to emphasize the enthusiasm for evangelism as a key factor in church growth, and why church image and improving it through positive social engagement and activism is also important. I recommend the work of Mara Einstein on the religious branding of the United Methodist Church (start with article titled: “The Evolution of Religious Branding”) to assist in this framing.

I disagree with the other reviewer regarding who the likely audience for the paper is. Given the complexity and novelty of SD/ABM methods, it seems unlikely to me that the audience will be church leaders and pastors. For that reason, I am not concerned with how the audience may perceive epidemiological and HIV model. I also do not see the usefulness of figure 7, it is nearly impossible to read.

Among my lesser concerns is that the new sections of the manuscript need a copy edit, including the typo to the submitted title.

If the authors are successful in framing the article, it may be a very well-rounded contribution.

7. PLOS authors have the option to publish the peer review history of their article (what does this mean?). If published, this will include your full peer review and any attached files.

Reviewer #1: No

Reviewer #2: No

---

## [Author Response · Author response to Decision Letter 1]

7 Apr 2021

Dear Dr. Siew Ann Cheong,

Thank you for inviting us to submit a revised draft of our manuscript entitled, "Sustainability of Religious Population" to PLOS ONE. We also appreciate the time and effort you and each of the reviewers have dedicated to providing insightful feedback on ways to strengthen our paper. Thus, it is with great pleasure that we resubmit our article for further consideration. We have incorporated changes that reflect the detailed suggestions you have provided. We also hope that our edits and the responses we provide below satisfactorily address all the issues and concerns you and the reviewers have noted.

Again, thank you for giving us the opportunity to strengthen our manuscript with your valuable comments and queries. We have worked hard to incorporate your feedback and hope that these revisions persuade you to accept our submission.

Sincerely,

Jae-Woo Lee

Editor suggestions:

Please address the remaining concerns of Reviewer 2. Also, 'Riligious Population' sounds awkward in the title, and 'Religious' is misspelled. Will the authors consider 'Religious Communities' instead?

RESPONSE: The authors change the title to “Sustainability of Religious Communities.”

Comments to the Author

Reviewer #1: Most of the points I commented on before have been improved and the paper is now very easy to read and understand.

In particular, it is easier to understand the allocation of topics, the position of the research in the timeline, and the correspondence and relationship with previous research in Korea.

We appreciate the useful review.

Reviewer #2: I appreciate the efforts the authors put into responding to my earlier comments. The removal of text in the first few pages and addition of other text throughout does improve the framing. The focus now seems more about the public image and functional purpose of the church in Korean society, and how improving these assists with membership sustainability by making the un-churched more open to joining the church. A key phrase in helping me understand this was on page 12: “Sustainable potential s reflects how amicable unbelievers are to the Christian faith, and integrates social factors in a society.” This contrasts to previous approaches, which were focused on conversion potential via the level of enthusiasm for evangelism among current members.

Unfortunately, the contribution of the sustainability potential still doesn’t become clear until nearly half-way through the paper. There is still a problem that sustainability is introduced in a way that many readers may find confusing. The introduction should be about the sustainability of church membership through many avenues including conversion efforts, not how religious values or beliefs are related to the field of sustainability. 

Response

We add a new paragraph explaining the sustainability potential in the introduction part as following:

“If the new believer's faith is not strengthened, they will lose their passion for faith after a certain period of time. New believers are more likely to lose networks with unbelieving friends, and they will have a hard time forming new networks within the church. Therefore, sustainability potential represents the degree to which unbelievers contact believers and form trust. It represents the degree of trade-off between what unbelievers lose as they gain faith and what they gain in social networks and new communities.”

The first two paragraphs should be written to highlight the way that past literature tends to emphasize the enthusiasm for evangelism as a key factor in church growth, and why church image and improving it through positive social engagement and activism is also important. I recommend the work of Mara Einstein on the religious branding of the United Methodist Church (start with article titled: “The Evolution of Religious Branding”) to assist in this framing.

Response:

Religious communities carry out a variety of methods with the aim of maintaining their own sustainability. The United Methodist Church used religious branding to enhance the public reputation and to change the position of the traditional denomination. It demonstrated the need of marketing tool for the sustainability of religious institutions (Einstein, 2011). Oosthuizen and Lategan (2015) pointed out that church leaders often had difficulty performing the basic management tasks they expected because leaders had insufficient management principles and skills as an organization. The authors argued that church and denomination leaders should participate in management principles and skills education for more effective, efficient and sustainable management. Körösvölgyi (2017) addressed that as the Christian population declined, it was necessary to look at the positive and negative experiences affecting the sustainability of the church amid changes in the world map such as global warming, urbanization, uneven distribution of population and wealth, and migration. The author argued that if the place of worship as sacred building equipped with an efficient architectural concept to be better suited to the community, it could have a positive impact on the sustainability of the church. 

Einstein M. The evolution of religious branding. Social compass. 2011 Sep;58(3):331-8.

Oosthuizen AJ, Lategan LO. " Managing the household of God": The contribution from management sciences to the sustainability of the church as an organization. Stellenbosch Theological Journal. 2015;1(2):551-68.

Körösvölgyi Z. The Sustainable Church: A New Way to Look at the Place of Worship. Periodica Polytechnica Architecture. 2017;48(2):93-100.

Church planting is a good strategy if the church is to be sustainable through conversion. Churches wishing to grow should invest equally in member satisfaction and recruitment (conversion). In addition, the ministry of welcoming new believers and discipling them should be balanced (Paas, 2018). De Wetter (2011) and Roozen (2008) described the health and growth of the church as the concept of church vitality. They insisted that the spiritual vitality of worship through enthusiastic, committed or active members could be the catalyst for membership growth. 

Young-Gi Chai (2010) showed through the small group ministry called as the house church in the Seoul Baptist Church of Houston that the growth and sustainability of the church could be caused by conversion of non-believers. By converting the traditional church into a small group-centered church, the church focused on converting non-believers and raising beginners as disciples. Churches participating in house church ministry reeducated pastors and lay leaders by holding seminars and conferences after establishing the institute named as The House Church Ministries with Chai. The Seoul Baptist Church of Houston also supported that other churches could sustain the small group ministry by providing short-term training. 

Church enthusiasts lead to successful evangelism by adding new members when they are primarily involved in the conversion of unbelievers. It is the sustainability of the church that drives growth through its ability to reproduce itself (Hayward, 2005; Medcalfe and Sharp, 2012).

Paas S. A Case Study of Church Growth by Church Planting in Germany: Are They Connected?. International Bulletin of Mission Research. 2018 Jan;42(1):40-54.

De Wetter D, Gochman I, Luss R, Sherwood R. UMC call to action: Vital congregations research project. UMC. org. 2010 Jun 28.

Roozen DA. American congregations 2008. Hartford Institute for Religion Research, Hartford Seminary; 2007. 

Chai YG, Chai D. A New Testament church in the 21st century: The house church. GLPI; 2010.

Hayward J. A general model of church growth and decline. Journal of Mathematical Sociology. 2005 Jul 1;29(3):177-207.

Medcalfe S, Sharp C. Enthusiasm and congregation growth: Evidence from the United Methodist Church. International Journal of Business and Social Science. 2012 May 1;3(9).

I disagree with the other reviewer regarding who the likely audience for the paper is. Given the complexity and novelty of SD/ABM methods, it seems unlikely to me that the audience will be church leaders and pastors. For that reason, I am not concerned with how the audience may perceive epidemiological and HIV model. I also do not see the usefulness of figure 7, it is nearly impossible to read.

Response:

We believe that the SD/ABM method contributes to scientific humanities research as well as statistical research. Since Fig. 7 is difficult to look at, we have only described the relevant content without Fig. 7.

Among my lesser concerns is that the new sections of the manuscript need a copy edit, including the typo to the submitted title.

If the authors are successful in framing the article, it may be a very well-rounded contribution.

Response:

We corrected the title typo and conducted a copy edit the modified and added parts.

---

## [Editor Report · Decision Letter 2]

13 Apr 2021

Sustainability of Religious Communities

PONE-D-20-30060R2

Dear Dr. Lee,

We’re pleased to inform you that your manuscript has been judged scientifically suitable for publication and will be formally accepted for publication once it meets all outstanding technical requirements.

Kind regards,

Siew Ann Cheong, Ph.D.

Academic Editor

PLOS ONE
---

## [Editor Report · Acceptance letter]

28 Apr 2021

PONE-D-20-30060R2 

Sustainability of Religious Communities 

Dear Dr. Lee:

I'm pleased to inform you that your manuscript has been deemed suitable for publication in PLOS ONE. Congratulations! Your manuscript is now with our production department. 

Kind regards, 

on behalf of

Dr. Siew Ann Cheong 

Academic Editor

PLOS ONE